# A virophage cross-species infection through mutant selection represses giant virus propagation, promoting host cell survival

Said Mougari [1,2 ✉], Nisrine Chelkha[1,2], Dehia Sahmi-Bounsiar[1,2], Fabrizio Di Pinto[1,2], Philippe Colson [1,2], Jonatas Abrahao[3 ✉] & Bernard La Scola[1,2 ✉]

Virus adaptation to new hosts is a major cause of infectious disease emergence. This mechanism has been intensively studied in the context of zoonotic virus spillover, due to its impact on global health. However, it remains unclear for virophages, parasites of giant viruses and potential regulators of microbial communities. Here, we present, for the first time to our knowledge, evidence of cross-species infection of a virophage. We demonstrated that challenging the native population of Guarani virophage with two previously unidentified giant viruses, previously nonpermissive to this virophage, allows the selection of a mutant genotype able to infect these giant viruses. We were able to characterize the potential genetic determinant (deletion) carried by the virophage with the expanded-host range. Our study also highlights the relevant biological impact of this host adaptation by demonstrating that coinfection with the mixture containing the mutant virophage abolishes giant virus production and rescues the host cell population from lysis.

[1] Unité MEPHI, Aix-Marseille Univ., Institut de Recherche pour le Développement (IRD), Assistance Publique - Hôpitaux de Marseille (AP-HM), 19-21 boulevard Jean Moulin, 13005 Marseille, France. [2] IHU Méditerranée Infection, 19-21 boulevard Jean Moulin, 13005 Marseille, France. [3] Laboratório de Vírus, Departamento de Microbiologia, Instituto de Ciências Biológicas, Universidade Federal de Minas Gerais, Belo Horizonte, Minas Gerais, Brazil postal code 31270-901. ✉email: saidmougari@ymail.com; jonatas.abrahao@gmail.com; bernard.la-scola@univ-amu.fr

Host range is defined as the number and nature of hosts in which a virus can multiply[1]. This parameter is predicted to play a determinant role in virus pathogenicity, maintenance in nature, and epidemiology[2]. Some viruses have evolved the capacity to expand their host range by biologically adapting to novel hosts. This phenomenon is known as host range expansion and requires the selection of specific mutations that enable a given viral species to replicate in a novel host[3]. In bacteriophages, such host range mutations target mostly the genes encoding the viral tail and baseplate involved in attachment to a receptor at the host cell surface[4,5].

Virophages are double-stranded (ds) DNA viruses that adopt a satellite-like lifestyle[6,7]. Studying the mechanism of host range in virophages is interesting because their replication requires the presence of two types of hosts, a giant virus and a cellular host of this giant virus, a protist. Sputnik, the first virophage to be isolated, was described in 2008[6]. Sputnik is able to infect mimiviruses of the three phylogenetic lineages (A, B, and C) whose members replicate in *Acanthamoeba* spp.[6,8]. In contrast, another virophage named Zamilon was found to exhibit a narrower host range, being able to replicate with mimiviruses from lineages B and C but not with those belonging to lineage A[9]. The resistance of lineage A mimiviruses to Zamilon has been linked to the presence of a defense system mediated by a multigene-containing operon named MIMIVIRE[10,11]. Although the MIMIVIRE mechanism of action is still controversial[12], the host range of Zamilon has been efficiently expanded to mimiviruses of lineage A by silencing the MIMIVIRE genes[10] and, more recently, by knocking out the putative canonical gene of the system using homologous recombination[13]. However, spontaneous host range expansion has never been described in virophages. Moreover, to date, Sputnik, Zamilon, and Guarani, a new Sputnik-like virophage recently isolated from a water sample collected in Brazil[14], were exclusively challenged with mimiviruses from the three lineages, A-C, but not with distant mimivirus relatives. The only virophage tested with a distant mimivirus was Mavirus, a virophage that replicates only in the marine flagellate *Cafeteria roenbergensis* coinfected with a distant mimivirus relative named CroV[15].

In this study, two recently isolated mimiviruses, known as Tupanvirus Deep Ocean and Tupanvirus Soda Lake, were challenged with virophages[16]. Tupanvirus Deep Ocean and Tupanvirus Soda Lake were found in ocean sediments and soda lake samples collected in Brazil, respectively. According to their distinct structural, genetic, and biologic features, these two giant viruses were classified as distant mimivirus relatives and have been proposed to be part of a new genus *Tupanvirus*[17,18]. Here, we report, for the first time to our knowledge, a preliminary identification of a mechanism of cross-species transmission of virophages to infect these giant viruses. We were able to characterize the genetic component involved in this process. We then conducted a comprehensive study to investigate the biological impact of this phenomenon through its effect on the previously unidentified virus host as well as on the host cell population. The involvement of the mutation in the expanded-host range of the virophage is discussed.

## Results

### First evidence of cross-species infection of a virophage.

We used distant mimivirus relatives[17], Tupanvirus Deep Ocean and Tupanvirus Soda Lake, as models to study the host range of virophages and their ability to replicate in amoebae infected with other mimiviruses than those belonging to the three lineages (A, B, and C) of the family *Mimiviridae*[19]. Three virophages characterized in earlier studies were assayed with these giant viruses,

including Guarani, Sputnik and Zamilon virophages[6,9,14]. These virophages were previously propagated in *Acanthamoeba castellanii* cells coinfected with mimiviruses and then purified. *A. castellanii* cells were inoculated with each Tupanvirus strain at a multiplicity of infection (MOI) of 10. The same MOI was used for each virophage. The replication of each virophage was then assessed by quantitative PCR (qPCR) at 0, 24 hours (h) and 48 h postinfection (p.i.). Finally, the increase in the amount of virophage DNA was calculated using the delta Ct method considering time points 0 and 48 h p.i. Sputnik and Zamilon were able to infect and replicate with both Tupanvirus Deep Ocean and Tupanvirus Soda Lake (Fig. 1a, b). Unexpectedly, it was not possible to detect Guarani replication with Tupanvirus Deep Ocean or Tupanvirus Soda Lake (Fig. 1a, b).

After the lysis of host cells coinfected with Guarani and Tupanvirus Deep Ocean or Tupanvirus Soda Lake, each culture supernatant was filtered through 0.22-μm-pore filters to remove giant virus particles. The obtained supernatants containing only Guarani particles were subsequently used to infect fresh *A. castellanii* cells simultaneously inoculated with Tupanvirus Deep Ocean or Tupanvirus Soda Lake at MOIs of 10. The replication of the virophage was then measured by qPCR. In these assays, Guarani remarkably infected and successfully replicated with both Tupanvirus Deep Ocean and Tupanvirus Soda Lake after only one passage with each virus (Fig. 1c, d). We interpreted this phenomenon as the result of a mechanism of mutant selection resulting in host range expansion that allowed Guarani to replicate with new viral hosts previously resistant to this virophage. A small fraction of the Guarani population in our stocks was likely composed of an emergent mutant able to infect Tupanvirus. We found that Guarani isolated from Tupanvirus Deep Ocean culture supernatant was able to replicate with Tupanvirus Soda Lake and *vice versa*. Taken together, these results present, to our knowledge, the first evidence of a cross-species virophage transmission between mimiviruses and their distant relatives.

### Potential host range mutation associated with the host acquisition.

In bacteriophages, host range expansion involves several molecular paths. Indeed, spontaneous or induced mutations in the long tail fiber gene have enabled some phages to infect new bacterial hosts[20–24]. To identify the genetic features responsible for the expanded-host range of Guarani, we sequenced the genome of this virophage obtained before and after the passage with each Tupanvirus. The Guarani genome consists of a dsDNA genome of 18,967 base pairs encoding 22 predicted genes very similar to Sputnik[14]. We found that the virophage isolated from Acanthamoeba polyphaga mimivirus (APMV) supernatant has no mutations in its genome compared to the original strain that we sequenced previously[14]. We therefore consider this strain to be the wild-type genotype of the virophage. Interestingly, genome analysis of Guarani isolated from Tupanvirus supernatant revealed the emergence of a new subpopulation of the virophage that shows a deletion in its genome. This deletion consists of a loss of an 81 nucleotide-long sequence located in ORF 8 that encodes a collagen-like protein (Supplementary Fig. 1, Supplementary Data 1, 2). The deletion has been detected in Guarani purified from both Tupanvirus Deep Ocean and Tupanvirus Soda Lake. We then aimed to confirm the results of the sequencing experiment by designing a PCR system that targets the deletion site, as shown in Fig. 2a. We were able to confirm the presence of two genotypes of Guarani isolated from Tupanvirus supernatant versus only one genotype in APMV supernatant (Fig. 2b). The band corresponding to each genotype detected in the Tupanvirus supernatant was recovered from the 2% agarose gel. Sanger

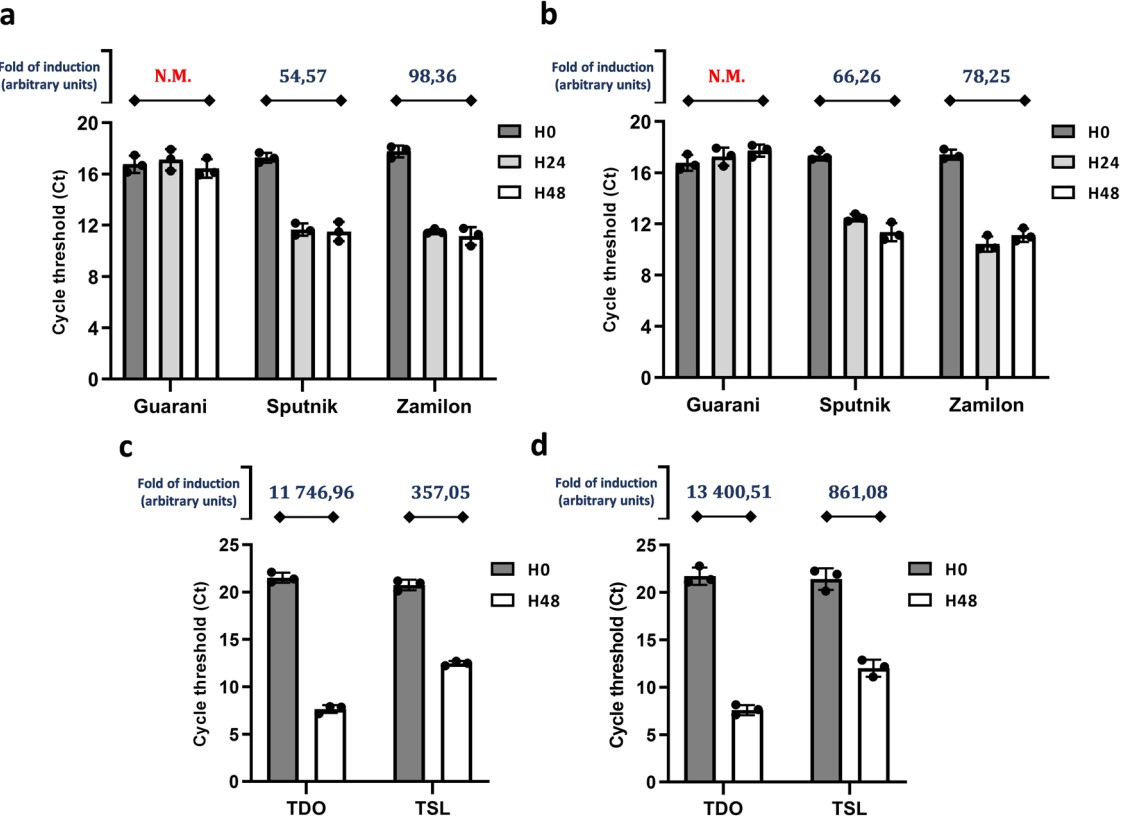

**Fig. 1 Expanded-host range of Guarani isolated from Tupanvirus cocultures. a**, **b** Histograms depicting the genome replication, during primo-cocultures, of Guarani, Zamilon and Sputnik in *Acanthamoeba castellanii* coinfected with Tupanviruses. **a** Replication of virophages with Tupanvirus Deep Ocean. **b** Replication of virophages with Tupanvirus Soda Lake. **c**, **d** Graphs depicting genome replication of Guarani isolated from Tupanvirus supernatants in *Acanthamoeba castellanii* coinfected with each Tupanvirus strain. **c** Guarani isolated from Tupanvirus Deep Ocean supernatant. **d** Guarani isolated from Tupanvirus Soda Lake supernatant. The DNA replication of each virophage at times 0, 24, and 48 h p.i. was measured by quantitative real-time PCR (for (**c**) and (**d**), only times 0 and 48 p.i. are shown here). The increase in the amount of virophage DNA (fold of induction) was then calculated using the delta Ct method considering the difference between the Ct values specific to each virophage at times H0 and H48. Amoebas infected only with each Tupanvirus was used as negative control. All the PCRs targeting these controls were negatives at times H0, H24, and H48. Error bars, standard deviation ($n = 3$ biologically independent experiments). N.M.: No multiplication.

sequencing was then performed on each band and confirmed the presence of the deletion (Supplementary Data 3). The same procedure was performed with Guarani cultivated with APMV and confirmed the presence of the wild-type genotype, but the mutant was not detected in this condition.

**Evolution of the mutant virophage and Tupanvirus during serial-passage experiments**. Virus adaptation to new hosts combines two steps: first, the introduction of mutations allowing the virus to infect new hosts and then maintenance of these mutations during spread within the host population[25]. With this in mind, we investigated the capacity of mutant Guarani to maintain the deletion during passage experiments using Tupanvirus Deep Ocean as a virus host. This Tupanvirus strain was selected because it enables the virophage to replicate with higher replication efficiency (Fig. 1c, d). We continuously cocultured Guarani 5 times in *A. castellanii* and subsequently characterized its progeny after each passage using our PCR system targeting the deletion site (Fig. 3a, Supplementary Fig. 2). This experiment has been started by a primo-coculture of Guarani wild-type with Tupanvirus followed by four subcultures. Each supernatant was taken to perform a new passage at time 48 h p.i. Samples for PCR targeting the deletion site were also collected at this time. The same procedure was carried out for APMV and Guarani wild-

type (control) (Fig. 3b, Supplementary Fig. 2). Passages with Tupanvirus promoted the propagation of the mutant genotype, rather than the wild-type strain (Fig. 3a). Clearly, according to the PCR product intensity, after several passages with Tupanvirus, the amount of mutant Guarani DNA evolved to decrease (Fig. 3a). Such a decrease was not observed in the control (Fig. 3b). These results are intriguing given the high replication efficiency of the mutant genotype observed with Tupanvirus Deep Ocean (Fig. 1c). At the same time, we noticed a progressive increase in the host cell population survival over the passages. Such an increase was not observed in the control, in which almost all the cells were lysed at 48 h p.i. but this parameter was not quantified in these conditions. These observations probably raise questions regarding the virulence of the mutant genotype toward its new virus host, Tupanvirus. Indeed, it is known from our previous study that Guarani impaired the infectivity of its viral host APMV, resulting in a decrease in amoebae lysis[14]. Therefore, inhibition of Tupanvirus could decrease mutant virophage propagation over passages. In addition, given that all supernatants in this experiment were collected at 48 h p.i., we speculate that the loss of mutant may also be caused by the delay in lysis of cells it infected. This may prevent the release of the virophage to infect fresh amoebas during subcultures, and this phenomenon gets worse as we go with passages causing mutant dilution. We then repeated the same experiment by adding fresh Tupanvirus at each

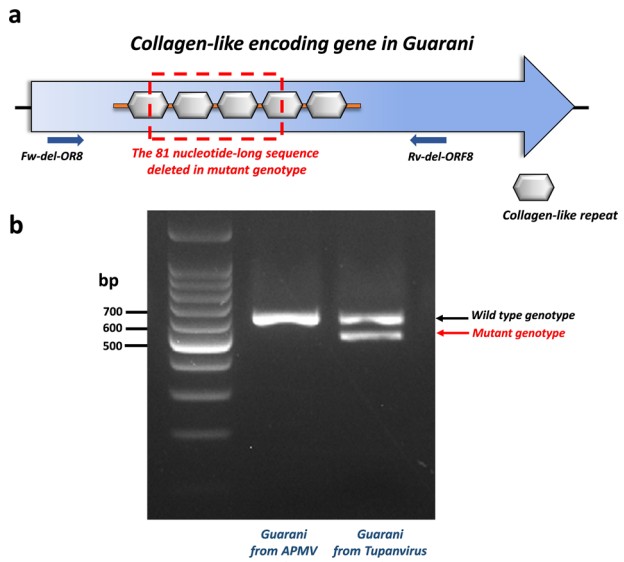

**Fig. 2 Characterization of the mutant genotype of Guarani isolated from Tupanvirus coculture. a** Schematic representation of the collagen-like gene in Guarani. This gene contains five collagen-like repeats of 27 nucleotides each. The mutant genotype shows a deletion of an 81 nucleotides sequence that affects four collagen-like repeats, from which two repeats are completely lost. **b** PCR characterization of the mutant genotype detected in Tupanvirus supernatant but not in APMV coculture. Only the product of PCR targeting Guarani isolated from Tupanvirus Deep Ocean is shown here. The primers used for this PCR system (Fw-del-OR8 and Rv-del-OR8) target the regions located around the deletion site. The genotype corresponding to each band was then confirmed by Sanger sequencing.

passage (Fig. 3c, Supplementary Fig. 2). We found that adding fresh giant virus allows the propagation and the maintenance of mutant Guarani. These results provide evidence that the deletion was the primary, but possibly not exclusive, determinant of host range expansion of Guarani. This finding also suggests that mutant Guarani may have a distinct virulent profile toward its novel virus host than the wild-type genotype toward APMV. To test this hypothesis, we first started by studying the evolution of Tupanvirus infectivity during a passage experiment in the presence of the mixture containing mutant Guarani. The Guarani used here was isolated from the second sub-coculture with Tupanvirus where the deletion was clearly observed. In order to allow the propagation of both viruses, coinfected cells were observed daily and each supernatant was used to perform a new passage after complete lysis of amoebas. In absence of total cell lysis, supernatants were collected at 5 days p.i. Total cell lysis was only observed in passages 1, 2, and 3. Then an increasing number of cells was observed in the supernatant during passages, but this parameter was not quantified in this experiment. Figure 3d shows Tupanvirus titers during a ten-passage experiment in the presence of the virophage measured by end-point dilution method. Prior to end-point dilution, the virus supernatant was submitted to heat treatment at 55 °C for 30 min[26]. This treatment efficiently inactivated the virophage without reducing the titer of viable giant virus particles (Supplementary Fig. 3). We found that infection with the mixture containing the mutant strain severely modifies the trajectory of Tupanvirus in the model. While the virus was found to increasing its titer over passages in the absence of the virophage, the presence of the latter caused a drastic decrease in virus propagation (Fig. 3d). Moreover, at the ninth passage, we were intrigued by the absence of cell lysis and any evident cytopathic effect on the host population. We then monitored the replication of Guarani virophage during this ten-

passage experiment. The procedure was also repeated by adding fresh Tupanvirus at each passage. At each passage, Guarani replication has been assessed by real-time PCR between supernatant inoculation and amoebae lysis or 5 days p.i. in absence of total cell lysis. The results were analyzed as described above. Figure 3e strongly supports that Tupanvirus decease affects Guarani replication. This replication was reduced to indetectable levels when Tupanvirus titer crossed the limit of under $10^2$ TCID$_{50}$/ml. Even at low levels of virophage replication (P4–P8), we observed further decrease in Tupanvirus titer. This suggests that low concentrated virophage could have more evident effect on Tupanvirus at low titer. Our results also show that adding fresh Tupanvirus in the system preserves the virophage propagation (Fig. 3e). Taken together, these observations probably indicate that the mutation not only enabled Guarani to replicate with Tupanvirus but also that the mutant virophage could be highly virulent to the point of inducing the eradication of the giant virus in the model.

**Virophage infection leads to Tupanvirus eradication.** To go further into this story, we aimed to investigate the effect of virophage mixture containing mutant Guarani on the replication of Tupanvirus during a one-step growth curve. To this end, *A. castellanii* cells were coinfected with Tupanvirus Deep Ocean and Guarani mixture, containing both wild-type and mutant genotype (according to PCR), at MOIs of 10. The Guarani used in this study was isolated and then purified from the second sub-coculture with Tupanvirus in which the mutation was detected. We first quantified the impact of virophage on genome replication of Tupanvirus at 48 h p.i. Figure 4a shows that the presence of mutant Guarani decreases the replication of Tupanvirus DNA by approximately 3-fold. This low inhibition rate appears relatively similar to what has been found between the wild-type strain and APMV[14]. So, probably the virophage inhibits giant virus propagation in another point of the cycle such as the morphogenesis. We therefore tested the effect of the virophage on the production of viable particles by quantifying the titer of Tupanvirus from 0 h to 72 h p.i. Our results reveal that the presence of mutant Guarani induces a severe decrease in the infectivity of Tupanvirus virions during the virus cycle (Fig. 4b). We found that in the absence of virophage, the giant virus was able to increase its titer by up to 500-fold during a one-step growth curve in amoebae. In contrast, infection with the mixture containing the mutant genotype prevented any increase in the virus titer. This inhibition (1000-fold reduction) is far higher than that of the wild-type toward APMV (10-fold reduction)[14]. However, at 72 h p.i., we still were able to detect the production of Tupanvirus infectious virions by end-point dilution.

To gain more insight into the virophage–giant virus interaction, amoebas were coinfected with Tupanvirus Deep Ocean and the mixture containing mutant Guarani at MOIs of 10 and prepared for transmission electron microscopy (TEM). Remarkably, TEM images revealed that presence of mutant Guarani induces a total inhibition of Tupanvirus particle morphogenesis (Figs. 4c–i). We scanned more than two hundred cells and quantified the number of amoebas, in which both virophage and giant virus progeny were observed (Fig. 4c). We considered only the amoebas, in which Guarani was detected to analyze the presence of Tupanvirus. Our results confirmed the absence of any simultaneous occurrence of Guarani and Tupanvirus virion production in all coinfected cells. The presence of the virophage was automatically associated with the absence of Tupanvirus virions (Figs. 4d–i). To characterize this phenomenon, we observed virus factories infected with the mixture containing mutant Guarani at serial stages of coinfection (Fig. 5). Figure 5a

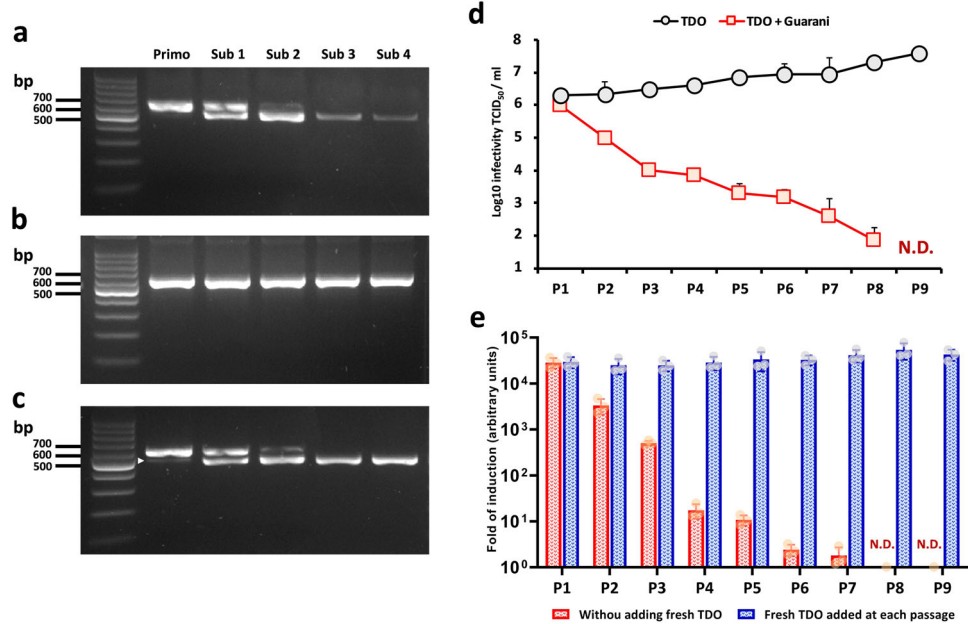

**Fig. 3 Selection of mutant Guarani during serial passage in Tupanvirus Deep Ocean. a** Selection of the Guarani mutant genotype coinfecting Tupanvirus during a 5-passage experiment could be visualized by PCR (lower band-arrow). **b** Maintenance of wild-type Guarani through passages with APMV (control). **c** The same experiment in (**a**) was repeated by adding fresh Tupanvirus at MOI of 10 at each passage revealing the selection and maintenance of mutant Guarani through passages. **d** Tupanvirus titer during 10 passages in the presence and absence of the mixture containing mutant virophage measured by the end-point method (the experiment was performed without adding fresh giant virus at each passage). **e** Histograms depicting the genome replication of Guarani coinfecting Tupanvirus during a serial-passage experiment performed by adding and without adding fresh giant virus at each passage. The DNA replication of the virophage between time 0 h p.i. and complete lysis of amoebas was measured by quantitative real-time PCR. In absence of total cell lysis, the supernatants were taken at 5 days p.i. The fold of induction corresponding to the increase in the amount of virophage DNA (represented here) was then calculated using the delta Ct method considering the difference between the Ct values specific to the virophage at time H0 and at the time of cell lysis (or 5 days p.i. in absence of complete amoebae lysis). Amoebas infected only with Tupanvirus was used as negative control. All the PCRs targeting these controls were negatives. Error bars, standard deviation ($n = 3$ biologically independent experiments). Primo: Primo-coculture wild-type and TDO. Sub: Sub-coculture. P: Passage. TDO: Tupanvirus Deep Ocean. N.D.: Not detectable.

shows that at 16 h p.i., the virus was able to produce mature virus factories, even in the presence of virophage. The first virophage progeny was observed at this step. Later in the cycle, at 20 h, 24 h, 30 h and 36 h p.i., respectively, the replication of the virophage increased progressively and caused remarkable damage to virus factories (Figs. 5b–e). The latter appeared disintegrated, and virophage progeny were actively emerging from each of their pieces. Virophage multiplication was clearly correlated with a progressive degradation of Tupanvirus viral factories. Isolation and characterization of these structures have previously shown that they are composed of an arsenal of virus-encoded proteins involved in virus replication and assembly[27]. Therefore, there is some overlap between our observations and previous studies reporting that virophages hijack their giant virus-encoded machinery to express and probably replicate their genomes[28,29]. In this study, we suggest that these small viruses obtain essential elements from the factories of giant viruses to propagate, causing their degradation.

However, it is still not clear how the giant virus was able to produce infectious virions at the end of its cycle (Fig. 4b), despite the presence of the virophage that is supposed to induce its elimination. We therefore quantified the rate of infected cells for each virus (Tupanvirus and Guarani). We found that all observed cells were successfully infected with Tupanvirus (100%). Although no giant virions have been produced in presence of the virophage, the presence of their volcanic-like virus factory in the cytoplasm of a given host cell was considered a sign of infection with the giant virus. On the other hand, virophage virions were observed in approximately 76% of host cells (152

cells from 200 cells observed). This finding means that up to 24% of amoebas were successfully infected only by Tupanvirus alone without the virophage. Two plausible explanations could justify this observation. First, the virophage suspension used here contained both mutant and wild-type genotypes; thus, the latter was not able to replicate in cells it infected. Second, the titration of virophage was carried out using qPCR which, in contrast to end-point dilution, targets both infectious and defective particles. Based on these results, we speculate that giant virus particles produced during coinfection experiments with Tupanvirus and mutant Guarani at MOIs of 10 were released from amoebas infected only by the giant virus (and probably the wild-type genotype). We therefore reduced the MOI of Tupanvirus from 0.01 to 1 and performed a dose–response experiment to virophage (Fig. 6). To improve the infection rate with virophage prior to coinfection, the virophage was incubated with Tupanvirus for 30 min at 30 °C to allow the formation of giant virus–virophage composite. Indeed, Sputnik-like virophages are thought to enter their host cell simultaneously with their associated giant virus, attaching to its capsid fibrils[30,31]. Figure 6c demonstrates that regardless of the MOI of Tupanvirus, higher virophage concentrations systematically reduced the titer of Tupanvirus to undetectable levels. In parallel with this result, Zamilon was not able to cause such inhibition for Tupanvirus (Fig. 6b), and simultaneous production of virophage and giant virus progenies was observed in coinfected cells (Supplementary Fig. 4). On the other hand, although the mixture containing mutant Guarani has caused a decrease in APMV titer, increasing MOI of this virophage was not able to completely prevent the

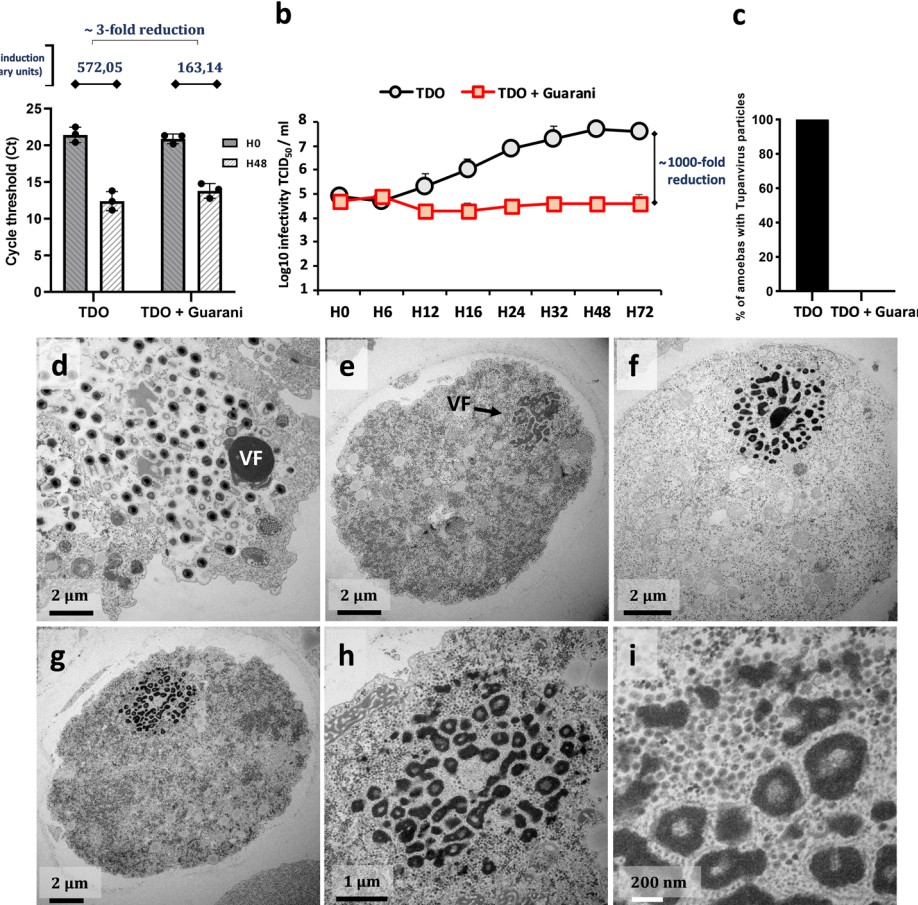

**Fig. 4 Characterization of the biological activity of the mixture containing mutant Guarani on Tupanvirus Deep Ocean replication cycle. a** Genome replication of Tupanvirus in the presence and absence of mutant virophage measured by quantitative real-time PCR. The Delta Ct method was used to analyze the increase in virus DNA for each condition between times H0 and H48 as described above. Error bars, standard deviation. **b** A one-step growth curve of Tupanvirus in the presence and absence of mutant Guarani showing that the virophage drastically reduces the ability of the virus to produce viable virions in coinfected amoebas. A slight increase in Tupanvirus titer was observed from H12 to H32, which is probably due to the increasing morphogenesis of the giant virus virions in cells coinfected by wild-type Guarani or infected only by Tupanvirus ($n = 3$ biologically independent experiments). **c–i** Transmission electron microscopy analyses show that presence of mutant virophage induces a total inhibition of Tupanvirus morphogenesis. **c** Percentage of infected amoebas in which Tupanvirus virions have been observed in the presence and absence of mutant virophage. Graph **c** was generated by analyzing the viral factories of 200 coinfected cells. Only the amoebas in which Guarani was detected were considered to analyze the presence of Tupanvirus. Error bars, standard deviation. **d–i** Transmission electron microscopy images of *A. castellanii* cells infected with Tupanvirus at 24 h p.i. **d** In the absence of virophage infection, the cell cytoplasm is fulfilled by mature viral particles. **e–i** The presence of mutant Guarani completely interrupts the production of Tupanvirus virions in coinfected cells, in which only the virophage progeny could be observed (**i**). (**g–i**) show the same cell with different zooms. TDO: Tupanvirus Deep Ocean. VF: Virus factory.

propagation of the virus (Fig. 6a). These results confirm that Tupanvirus was unable to establish a productive infection in host cells that are simultaneously coinfected by the mutant genotype of Guarani.

**Host cell population survives Tupanvirus infection in the presence of the mutant virophage**. The next step was to investigate how the mutant Guarani could manipulate the stability of the host cell population. This effect was noticed during the experiments in Fig. 3 but was not quantified precisely. To this end, *A. castellanii* cells in PYG (Peptone, Yeast extract, Glucose) medium were simultaneously inoculated with Tupanvirus at several MOIs (from 0.01 to 1) and the mixture containing mutant Guarani at a higher MOI (MOI = 10). The concentration of the host cell population was then monitored by microscopy count from 0 h to 96 h p.i. In the absence of Guarani, Tupanvirus infection led to a dynamic decrease in cell density until causing a total cell lysis at 96 h approximatively (Fig. 6d). Interestingly, at

low MOIs of Tupanvirus (0.1 and 0.01), coinfection with Guarani was able to stop the propagation of the virus and thus rescued the host cell population from lysis (Fig. 6d). Even in the presence of the virophage, we observed a cytopathic effect (rounded cells) and detected cell lysis with Tupanvirus at an MOI of 1. This finding may be observed because infection by Tupanvirus inevitably causes cell lysis, even when no virions could be produced (Fig. 6c). This observation is similar to what has been described for CroV[32]. However, by reducing the virus MOI, we observed that neighboring cells did not seem to be affected after lysis of infected cells, most likely because the virophage was able to efficiently neutralize the virus in coinfected cells.

In parallel, we also quantified the replication of Guarani at different MOIs of Tupanvirus. Figure 6e shows that virophage replication clearly depends on Tupanvirus MOI. We found that reducing the MOI of the giant virus impaired Guarani replication. This could be due to the low number of coinfected amoebas but also to the absence of Tupanvirus spread in presence of Guarani.

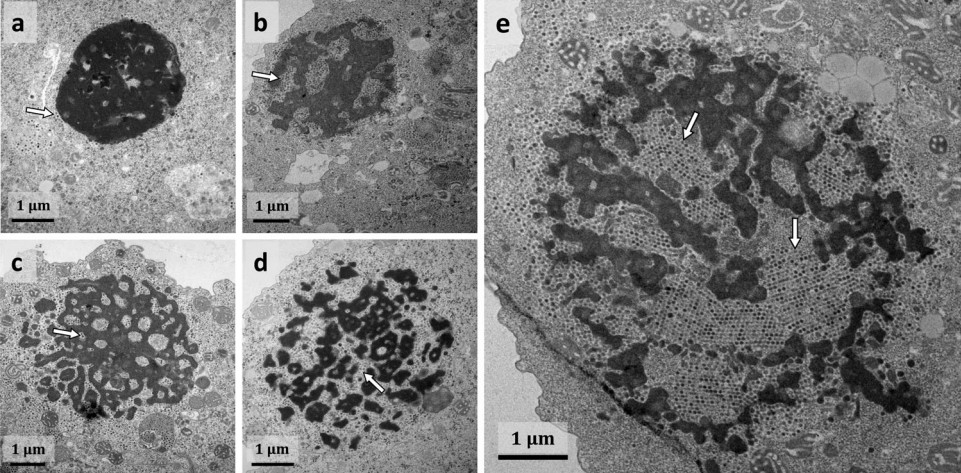

**Fig. 5 Virophage replication is associated with a progressive degradation of Tupanvirus viral factories.** Transmission electron microscopy images of Tupanvirus viral factories infected with mixture containing mutant Guarani at different stages of infection. **a** The first virophage progeny could be observed at 16 h p.i. (arrow), at this stage, the virus factories seem to be still intact. **b–e** Later in the cycle, at 20 h, 24 h, 30 h and 36 h p.i., respectively, the replication of the virophage (arrows) causes a remarkable fragmentation of the virus factories and seems to be correlated with their progressive degradation, affecting Tupanvirus particle morphogenesis.

**Analysis of mutant Guarani fitness with giant viruses belonging to the three phylogenetic lineages of the family Mimiviridae.** We then analyzed whether host acquisition of Guarani was associated with a fitness trade-off with the prototype isolate of genus *Mimivirus*, APMV, which has been used beforehand to propagate the wild-type genotype of Guarani[14,33]. Indeed, experimental evolution studies for other viruses have shown that increasing the virus fitness in one host could result in a fitness penalty in another host[1]. We compared the replication efficiency of Guarani before and after its passages with Tupanvirus using APMV as a giant virus host. Only the fitness of Guarani isolated from Tupanvirus Deep Ocean was investigated here because this virophage was the only one to be well characterized and enriched by the passages. The mutant used here was isolated after four sub-cocultures with Tupanvirus (Fig. 3a) and presents the same deletion in ORF 8 as mutation in its genome. Figure 7 shows that even after the acquisition of new virus hosts, the virophage maintained its capacity to replicate with APMV. In addition, the replication efficiency of Guarani isolated from Tupanvirus supernatant with APMV was similar to that of the original strain of Guarani propagated with APMV (Student's *t* test, *p* > 0.67). Similar results were obtained with Moumouvirus and Megavirus Courdo 11, which are mimiviruses from lineages B and C, respectively[34,35]. We also noticed that mutant Guarani fitness with Tupanvirus seems significantly higher than its replication with APMV or than that of the wild-type with this giant virus (Figs. 1c, d, Fig. 7) (Student's *t* test, *p* < 0.0005). One might suppose that this could be related to the adaptation of Tupanvirus transcription machinery to mutant Guarani genome. However, only extensive bioinformatic analyses could verify this hypothesis and provides further explanations.

**Tupanvirus-induced host-ribosomal shutdown prevents virophage infection.** Tupanvirus is able to trigger a cytotoxic profile associated with a shutdown of ribosomal RNA (rRNA) in host and nonhost organisms[16]. This profile probably allows the virus to modulate nonhost predator organisms to increase viral survival chances in nature. In host organisms, such as *A. castellanii*, this phenomenon is mainly observed at high MOIs like MOI of 100. In these conditions, the virus causes the cytotoxicity of its host cell without replicating[16]. Here, we investigated how this cytotoxicity could modulate virophage parasitism and notably infection by the

mutant genotype of Guarani. To the best of our knowledge, such a study has never been conducted. *A. castellanii* cells were coinfected with Tupanvirus at an MOI of 100 and each virophage (Sputnik, Zamilon or the mixture containing mutant Guarani) at MOIs of 10. Moumouvirus, a lineage B mimivirus, was used as a control because it allows the replication of all virophages used in this study. First, at 9 h p.i., $5 \times 10^5$ cells were collected to check the impact of Guarani on ribosomal shutdown. We found that in contrast to the controls, irrespective of mutant Guarani presence, Tupanvirus was able to induce a severe shutdown in host-ribosomal RNA abundance (Fig. 8a). We then measured the replication of each virophage by qPCR and calculated their DNA amount increase using the delta Ct method considering times 0 and 48 p.i. All virophages were able to replicate with Moumouvirus at the MOI of 100 (Fig. 8b). In contrast, all of the virophages, including Guarani, failed to infect Tupanvirus at high MOI (Fig. 8c). These results suggest that the cytotoxic profile of Tupanvirus allowed it to escape virophage infection and, notably, the eradication that could be caused by Guarani.

## Discussion

Evolution is an inevitable path for living organisms to adapt to changes in their ecosystem and to explore new environmental niches. In microorganisms, the evolutionary process seems to be driven by two major factors: genome plasticity and selection[36]. In this study, we report the first description, to our knowledge, of a virophage cross-species infection, which is mediated by nucleotide deletion and selection of mutant.

The first question concerns the origin of the deletion. While both Illumina sequencing and PCR failed to detect the mutant genotype in the initial isolate of the virophage, several studies have shown that host range mutations usually exist in the viral population before contact with the new host as part of the virus's genetic diversity[3,37]. Moreover, the emergence of spontaneous mutations associated with host adaptation in other DNA viruses usually requires more than one passage[38]. Therefore, the most credible explanation is that the mutant genotype was already present in the Guarani population that seemed not able to replicate with Tupanvirus. The capacity of the expanded mutant to infect mimiviruses of group A, B and C, at the same level as the wild-type, strongly suggests that this mutant could also propagate together with the wild-type during the infection with APMV used

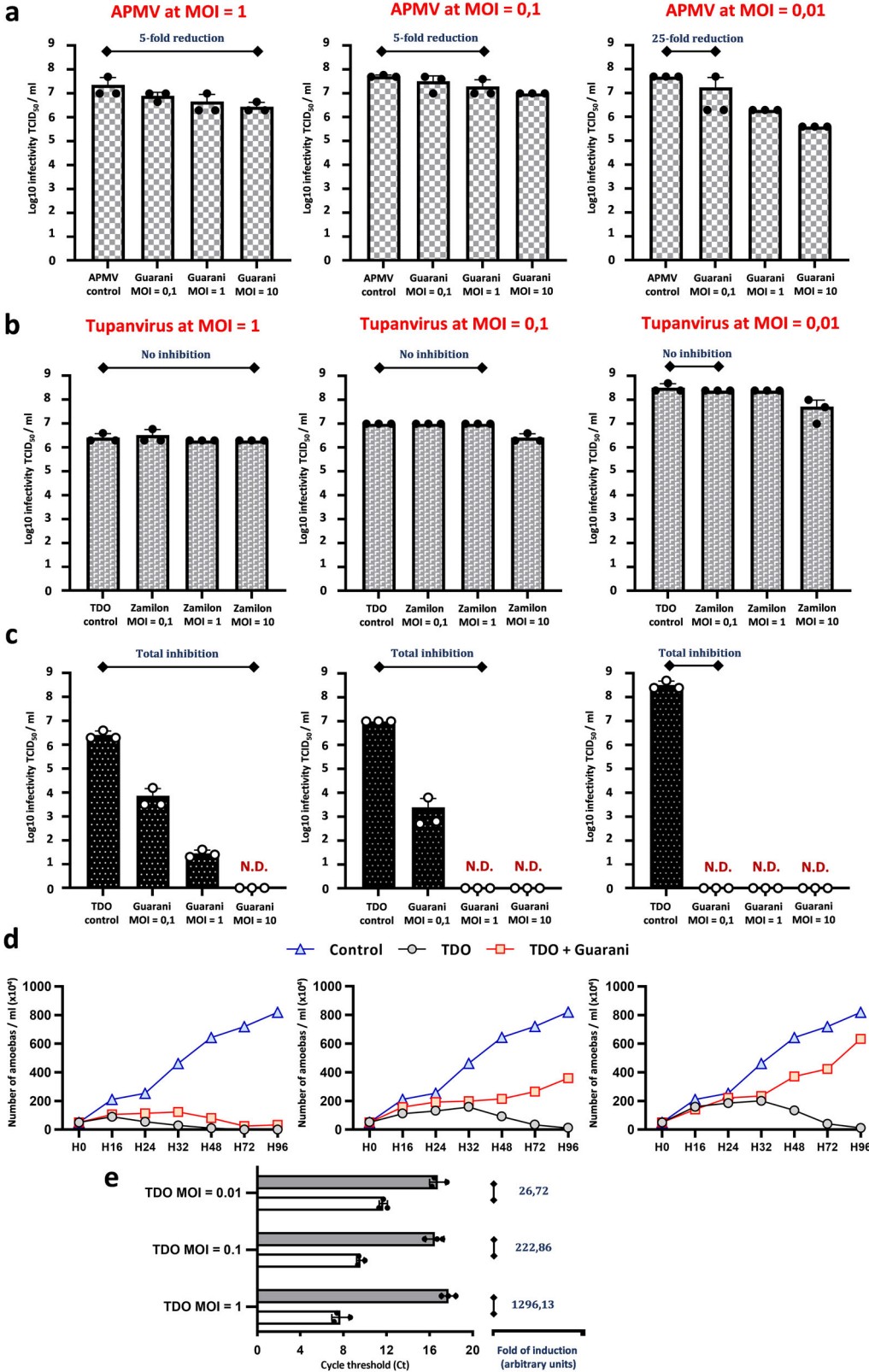

to generate the first Guarani stock. However, its concentration was probably under the limit of detection of our systems, most likely due to the dominance of the wild-type genotype in the mixture. Then, Tupanvirus allowed the selection of the mutant in coinfected cells. The presence of a very low amount of the mutant virophage after the primo-coculture with Tupanvirus probably

supports this hypothesis (Fig. 3c, arrow). This finding also suggests that the virophage has replicated even in the primo coinfection with Tupanvirus. The efficiency of this replication was probably at levels too low to be detected by real-time PCR but high enough to expand the virophage population, enabling its multiplication in subcultures.

**Fig. 6 Virophage halts Tupanvirus spread and protects the host cell (amoebas) population from lysis. a–c** Dose-response of giant virus to virophage. *A. castellanii* cells were coinfected with APMV (**a**) or Tupanvirus Deep Ocean (**b**, **c**) at different MOIs (1, 0.1 and 0.01) and increasing MOIs (0.1, 1 and 10) of the virophage mixture containing mutant Guarani (**a**, **c**) or Zamilon (**b**). The Tupanvirus titer in the supernatant was then measured by end-point dilution after complete lysis of cells or 5 days p.i. in the absence of cell lysis. **c** Higher concentrations of virophage cause a systematic fall in Tupanvirus titer to undetectable levels. We interpreted these observations as the results of a phenomenon of elimination induced by virophage. This phenomenon was not observed when Tupanvirus was challenged with Zamilon (**b**) nor when APMV was challenged with the virophage mixture that contains mutant Guarani (**a**). **d** Analyses of host cell survival in presence and absence of mutant Guarani infection. *A. castellanii* cells in PYG medium were coinfected with Tupanvirus Deep Ocean at different MOIs (1, 0.1 and 0.01) and virophage mixture of wild-type and mutant Guarani at an MOI of 10. Cell densities were then monitored by microscopy count from 0 h to 96 h p.i. Uninfected amoebas were used as controls. In the absence of virophage, regardless of Tupanvirus MOI, the virus lyses the entire host cell population at 96 h p.i. approximately. The presence of virophage dramatically modifies the lysis of amoebas at low MOIs of the Tupanvirus. The virophage protects a portion of host cells from lysis at MOIs of 0.1 and 0.01 of Tupanvirus. At an MOI of 0.01 of Tupanvirus, the phenotype of the amoebae population seems similar to that of uninfected cells in presence of mutant Guarani. **e** Histograms depicting the genome replication of Guarani with Tupanvirus Deep Ocean at different MOIs (1, 0.1 and 0.01). DNA quantification was done by Real-time PCR and analyzed as described above. Figures, in (**b–d**), are listed according to the MOI of Tupanvirus. Error bars, standard deviation (n = 3 biologically independent experiments). N.D.: Not detectable. TDO: Tupanvirus Deep Ocean.

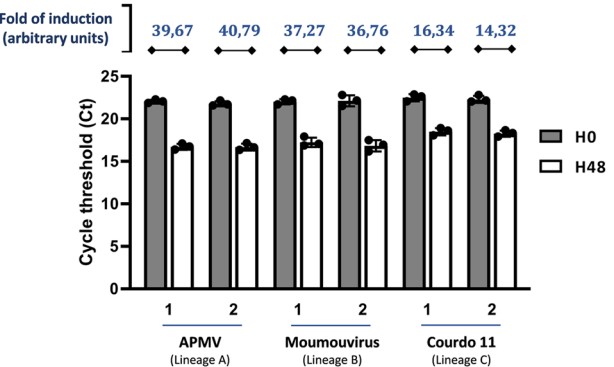

**Fig. 7 Comparative analyses of mutant Guarani fitness with giant viruses from the three phylogenetic clades of the family *Mimiviridae*.** Graph depicting genome replication of Guarani isolated from sub-coculture 4 with APMV (1) or Tupanvirus Deep Ocean (2) supernatants with different mimiviruses belonging to the three phylogenetic lineages A, B and C. The DNA replication of the virophage at times 0 and 48 h p.i. was measured by quantitative real-time PCR. The increase in the amount of virophage DNA (fold of induction) was then calculated using the delta Ct method considering the difference between the Ct value specific to virophage at times H0 and H48. Amoebas infected only with each giant virus was used as negative control. All the PCRs targeting these controls were negatives at times H0 and H48. Error bars, standard deviation (n = 3 biologically independent experiments).

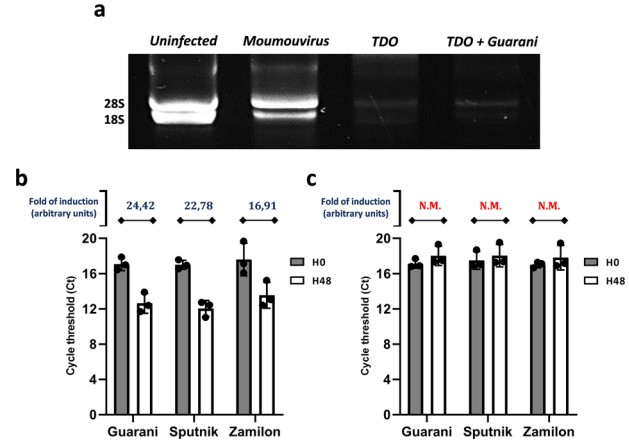

**Fig. 8 Virophage infection and rRNA shutdown induced by Tupanvirus.** *A. castellanii* cells were coinfected with Tupanvirus Deep Ocean at an MOI of 100 and each virophage at MOIs of 10. **a** Electrophoresis gel showing the ribosomal RNA profile (18S and 28S) from *A. castellanii* in the presence and absence of Guarani. Moumouvirus was used as a giant virus control. Tupanvirus has remarkably induced a severe shutdown in rRNA even in the presence of virophage. **b** All virophages replicate with Moumouvirus at an MOI of 100. **c** All virophages failed to replicate in amoebas coinfected with Tupanvirus at an MOI of 100. Error bars, standard deviation (n = 3 biologically independent experiments). TDO: Tupanvirus Deep Ocean.

Several studies have reported the occurrence of spontaneous large genomic deletions in dsDNA viruses after several passages. This includes the giant Mimivirus, poxviruses, African swine fever virus (ASFV), and chlorella viruses[39–42]. Spontaneous in-frame deletions have also been reported for several other viruses like influenza virus and severe acute respiratory syndrome coronavirus (SARS-CoV). Depending on the affected genes, these mutations could have a beneficial or deleterious impact on the viral fitness[43,44].

In our case, the deletion affected a part of a gene. ORF 8 in Guarani is 933 base pairs in length and contains 310 amino acids[14]. The same gene was described in all Sputnik strains (with 100% amino acid identity) and was predicted to be involved in protein–protein interactions with giant viruses within factories[6,8]. This gene shows a remarkable repetitive pattern. Five repeats of 27 nucleotides could be detected. The deletion affects 4 repeats, of which 2 repeats are completely lost (Fig. 2a). Our PCR system targeting the deletion site confirmed the presence of a new sub-population in Sputnik that shows the same deletion as Guarani

even before contact with Tupanvirus. This finding probably explains the capacity of Sputnik to replicate with this virus in primo-coculture (Supplementary Fig. 5).

Although it is difficult to predict the exact function that this gene could play during virophage infection, one plausible scenario is that ORF8 is implicated in the recognition and attachment to Tupanvirus proteins. Indeed, collagen-like motifs play a potential role in the attachment of phages to target bacteria. These motifs have been described in tail fiber proteins of several bacteriophages, and it is known from previous studies that mutations in this region are a crucial step in determining the acquisition of new hosts for these viruses[45,46]. On the other hand, collagen-like repeats have also been suggested to represent recombination hotspots for bacteriophages[45]. Interestingly, a spontaneous deletion of a region containing a collagen-like repeat has been described for the temperate *Streptococcus thermophilus* phage phi SFi21 after serial passage in bacteria. The mutant phage was remarkably unable to lysogenize its host cells[47]. It is tempting to link these observations to the study of Desnues et al[48]. The authors found that in Sputnik 2, the collagen-like gene also plays

a role in recombination spot that allows the virophage to integrate itself into the Lentillevirus genome as provirophage. Therefore, the deletion in ORF8 could also involve a process of integrating Guarani into the genome of Tupanvirus that could allow the virophage to replicate. However, an experimental setting to confirm such a hypothesis is challenging and is not currently possible to set up. Nevertheless, these data allow us to propose that the collagen-like gene contributes to the flexible gene content of virophages, giving them an advantage in their host-parasite interaction with giant viruses.

The second major finding of our study is that adapted virophage was highly virulent enough to induce the elimination of its associated virus. This observation reminds what has been described for the *S. thermophilus* phage phi Sfi21, for which a spontaneous deletion of a collagen-like repeat containing-region has transformed this temperate phage to a pure lytic phage[47]. However, the unique virophage control used in our experiments was Zamilon, and according to a previous study, this virophage does not seem to be virulent for its associated giant virus[9]. Hence, an alternative scenario is that the eradication of Tupanvirus was caused by its hypersensitivity to virulent virophage; thus, any other virulent virophage might provoke the same effect. Otherwise, apart from Guarani, the only known virulent acanthamoeba virophage is Sputnik. However, Sputnik could probably not be considered as an appropriate control because a subpopulation carrying the same mutation (as Guarani) was detected in this virophage.

We demonstrated here that coinfection with mixture containing mutant Guarani confers total protection to neighboring cells by abolishing the production of Tupanvirus virions in coinfected cells. This finding, as well as the findings of previous studies, supports the hypothesis of a protective role of virophages toward their host cells. Indeed, Fischer et al. found that the virophage Mavirus has the ability to integrate its genome into that of *C. roenbergensis* cells and remains latent. Superinfection with the giant CroV triggers the expression of the provirophage, enabling virophage replication. Mavirus then acts as an efficient inhibitor of CroV by preventing its spread in neighboring cells[32,49,50]. The main advantage of integrating virophages is probably that the host cell carries a permanent antiviral weapon in its genome. However, random integration of foreign genetic elements might have considerable impacts on the cell. These repercussions might range from altering its survival to affecting its evolutionary course[51]. In this context, infection with highly virulent virophages, such as mutant Guarani, may be more advantageous for amoebae than integration of Mavirus for *C. roenbergensis*. However, the fate of the virophage is also different between these two varieties of tripartite interactions. While the Mavirus genome is efficiently preserved within the *C. roenbergensis* genome, extinction of their associated giant virus will most likely cause the extinction of highly virulent nonintegrating virophages. We observed here that virophage virions contained in the mixture of wild-type and mutant were no longer viable after two months of incubation at room temperature (25 °C) and five months at 4 °C, approximatively (Supplementary Fig. 6). This finding suggests that the protection conferred to amoebae in their original habitats is probably temporary and remains dependent on viability of the virophage. Overall, this observation and that of Fischer et al., with integration of Mavirus (the predator of the predator) to *C. roenbergensis* (the prey), shows that interrelations in nature between these microorganisms are more complex than the extended Lotka–Volterra model of host–Organic Lake phycodnavirus (OLPV)–Organic Lake virophage (OLV) population dynamics proposed by Yau et al[52].

Our results also indicate that the only way for Tupanvirus to survive virophage infection is to trigger its cytotoxic profile. The incapacity of virophages to propagate in this condition could be

related to the absence of Tupanvirus replication but also to the shutdown of the host rRNA. This effect requires a high MOI for host organisms, and we have no evidence about the existence of such MOIs in nature. Overall, this observation and the data presented above allows us to present a new model of giant virus–virophage interaction (Fig. 9) in which the same giant virus could behave differently to virophage infection according to different parameters related to the standing genetic diversity of virophages but also to its concentration in the ecosystem.

To the best of our knowledge, our study was the first to provide evidence of virophage abilities to expand their host range to infect new giant viruses. This study also highlighted a relevant impact of this host adaptation on giant virus and virophage replication and on lysis of their host cells. Thus, our results help to elucidate the parasitic lifestyle of virophages and their ecological influence on giant virus and protist populations.

## Methods

**Tupanvirus production.** *Acanthamoeba castellanii* cells (ATCC 30010) cultivated in PYG (Peptone, Yeast extract, Glucose) medium were used to produce Tupanvirus Deep Ocean and Tupanvirus Soda Lake. Suspensions containing $7\times10^6$ cells plated in T175 flasks (Thermo Fisher Scientific, USA) were inoculated with each virus at an MOI of 0.02 and incubated at 30 °C. After complete lysis of the cells, each virus supernatant was collected and centrifuged at 1000$g$. The obtained supernatant was then filtered through a 0.8-μm membrane to remove amoeba debris. Each viral pellet was submitted to three cycles of wash with Page's modified Neff's amoeba saline (PAS) by ultracentrifugation at 14,000$g$ for 1 h. Finally, each virus was purified through ultracentrifugation across a 25% sucrose cushion at 14,000$g$ for 1 h.

**Virophage production.** APMV was used to propagate the Sputnik and Guarani virophages. To produce the Zamilon virophage, Megavirus Courdo 11 was used as a giant virus host. *A. castellanii* trophozoites at a concentration of $5\times10^5$ cell/ml in PYG were inoculated with each virus at an MOI of 10 and each virophage. After lysis of the cells, the supernatant containing both virus and virophage particles was centrifuged at 10,000$g$ for 10 min and then filtered through 0.8-, 0.45- and 0.22-μm-pore filters to remove giant virus particles and residual amoebas. The virophage particles were concentrated by ultracentrifugation at 100 000$g$ for 2 h, and the pellet was resuspended with PAS. Each virophage was then purified through ultracentrifugation at 100,000$g$ for 2 h across a 15% sucrose cushion. A pure highly concentrated suspension was finally obtained for each virophage, in which the absence of mimivirus particles was confirmed by negative staining electron microscopy and inoculation in amoebas.

Specific PCR systems (Table 1) to discard cross-contamination were also performed after the production of each virophage. The presence of Zamilon in Sputnik stock was negative and vice versa. We also designed a PCR system targeting ORF19, which is specific to Guarani and confirmed, as expected, that we were detecting Guarani. This system was also used to discard the presence of Guarani in Sputnik and Zamilon stocks.

**Host range studies.** *A. castellanii* cells, the cellular support of the system, were resuspended three times in PAS. Ten milliliters of rinsed amoeba at $5\times10^5$ cell/ml were simultaneously inoculated with Tupanvirus Deep Ocean or Tupanvirus Soda Lake and each virophage at MOIs of 10. The cocultures were incubated for 1 h at 30 °C, and then extracellular giant virus and virophage particles were eliminated by three successive rounds of centrifugation and resuspension in PAS (1000$g$ for 10 min). The cocultures were then submitted to a second incubation at 30 °C. This time point was defined as H0. Each Tupanvirus strain was separately incubated with amoeba in the absence of virophage to serve as a negative control. At time points 0, 24 and 48 h p.i., a 200 μl aliquot of each coculture was collected from the supernatant for real-time PCR targeting the Major capsid protein (MCP) gene of each virophage (Table 1).

This experiment was first done for Sputnik and Zamilon (separately) and then few months later for Guarani when this virophage was isolated (and immediately after Guarani genome sequencing). Then, the results of coinfections were reproduced using Tupanviruses and the three virophages in the same experiment and in triplicate (shown in Fig. 1a and Fig. 2b). These experiments have been performed very delicately to avoid contaminations between virophage stocks.

To study the host range expansion of Guarani, the supernatant, obtained after lysis of the host cells coinfected with Guarani and Tupanvirus Deep Ocean or Tupanvirus Soda Lake, was filtrated through a 0.22-μm membrane to remove Tupanvirus particles. One hundred microliters of the filtrate containing only Guarani particles was subsequently used to infect fresh *A. castellanii* cells in PAS simultaneously inoculated with Tupanvirus Deep Ocean or Tupanvirus Soda Lake at MOIs of 10. The replication of the virophage was then quantified as described above. The experiment was carried out three times independently in duplicate.

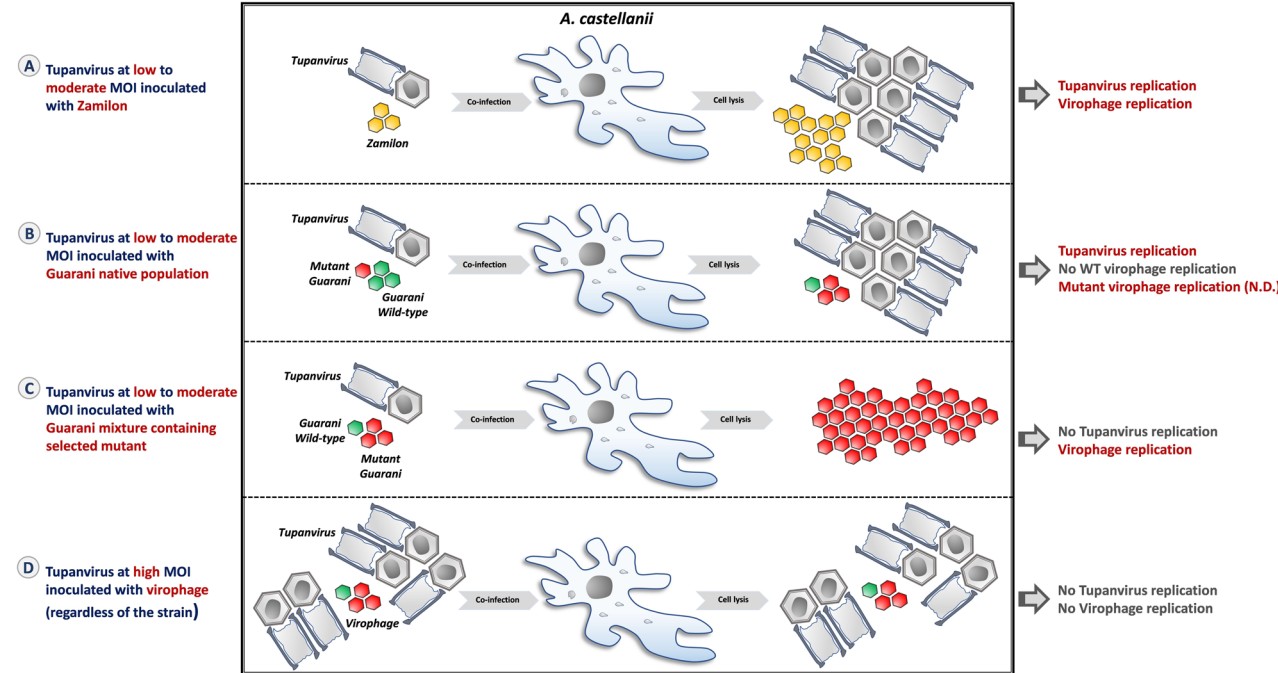

**Fig. 9 Scheme summarizing the different profiles of interaction between Tupanvirus and virophages described in our study. a** Coinfection of *A. castellanii* with Zamilon and Tupanvirus at low to moderate MOI (MOI ≤ 10) leads to replication of both Tupanvirus and virophage. **b** Guarani wild-type (WT) is not able to replicate with Tupanvirus in primo coinfection, but this first passage allows the selection of a mutant genotype adapted to the virus. **c** The mixture containing mutant Guarani is able to propagate with Tupanvirus and seems to be highly virulent, leading to the abolition of Tupanvirus morphogenesis. **d** Tupanvirus at high MOI (MOI ≥ 100) triggers its cytotoxic profile in host cells. Although the virus cannot replicate at this MOI, this feature allows it to prevent virophage parasitism. N.D.: Not detected.

### Table 1 Primers used for host range studies, mutant Guarani detection and characterization.

| Primers | Sequences | Tm | Target | Purpose |
|---|---|---|---|---|
| Fw1-Guarani | 5'- GAGATGCTGATGGAGCCAAT -3' | 59 °C | MCP gene | Host range studies |
| Rv1-Guarani | 5'- CATCCCACAAGAAAGGAGGA -3' | 59 °C | MCP gene | |
| Fw2-Guarani | 5'- TGGCAGCAGTTCAAGGTAAA-3' | 59 °C | ORF 19 | Control stocks |
| Rv2-Guarani | 5'- CCTGCTGCTAATTCATCAAATGGA-3' | 59 °C | ORF 19 | |
| Fw-Sputnik | 5'- GAGATGCTGATGGAGCCAAT -3' | 59 °C | MCP gene | Host range studies |
| Rv-Sputnik | 5'- CATCCCACAAGAAAGGAGGA -3' | 59 °C | MCP gene | |
| Fw-Zamilon | 5'- GGGATGAACATCAAGCTGGT -3' | 59 °C | MCP gene | Host range studies |
| Rv-Zamilon | 5'- GGGTTGTTGGAAGCTGACAT -3' | 59 °C | MCP gene | |
| Fw-del-ORF8 | 5'- AAGGTGATTCCGGAACTGATGG -3' | 60 °C | Collagen-like gene | Mutant genotype detection |
| Rv-del-ORF8 | 5'- AATTCCTGCGGTACTTGCTGTA -3' | 60 °C | Collagen-like gene | |
| Fw-TDO-MCP | 5'- GATGTGCTTGGACCTTCGGA -3' | 60 °C | MCP gene | Inhibition assays |
| Rv-TDO-MCP | 5'- AAGCGCGGAATTCTAGCTGT -3' | 60 °C | MCP gene | |

*TDO* Tupanvirus Deep Ocean.

The PCR system targeting ORF19 specific to Guarani and not Sputnik has also been used to quantify the replication of this virophage in primo-coculture with APMV, Tupanviruses and then during subcultures with Tupanviruses. These results confirmed our previous results regarding the host specificity of Tupanviruses to Guarani and the capacity of the latter to propagate in subcultures with these giant viruses. Only the results with PCR targeting MCP are shown in Fig. 1 to allows comparison with Sputnik.

**Real-time PCR**. DNA extraction and PCRs were performed as described by Mougari et al[14]. All the primers used here are listed in Table 1. Virophage replication with each Tupanvirus strain was calculated by the ΔCt method, considering the difference between times 0 and 48 h p.i.

**Detection and characterization of the mutant genotype**
*Selection of the mutant Guarani.* Two T175 flasks (Thermo Fisher Scientific, USA) containing each of 20 million *A. castellanii* cells in PYG medium were inoculated with Tupanvirus Deep Ocean and Guarani wild-type (propagated with APMV) at MOIs of 10. The coculture was incubated at 30 °C. After complete lysis of the cells,

the virus–virophage supernatant collected from each flask was used to infect 10 more T175 flasks, and the cocultures were incubated at 30 °C. Approximately 1 L of the virus–virophage supernatant was collected from all cocultures. The virophage particles were then purified as described above and subsequently submitted to genome sequencing. The same procedure was repeated for Tupanvirus Soda Lake and APMV (control).

*Genome sequencing and analyses.* To investigate whether the Guarani obtained from Tupanvirus Deep Ocean and Tupanvirus Soda Lake were genetically different from that propagated with APMV, the virophage cultivated with each virus was submitted to genome sequencing with Illumina MiSeq. The genome of each virophage was assembled through Spades software (default parameters) and manual finishing, and analyzed as previously described by Mougari et al.[14,53]. Comparative genomic analysis and genome alignment were then conducted using Muscle software and BLASTn alignment (Basic Local Alignment Search Tool)[54].

*Standard PCR and Sanger sequencing.* A specific PCR system targeting the collagen-like protein (ORF8) was designed and performed on the DNA extracted from supernatant containing the mutant genotype to confirm the occurrence of

deletion of the 81 nucleotide-long sequence in this strain (Fig. 2a). The details of the primers used are listed in Table 1, and the PCR product was visualized on an agarose 2% gel using SYBR safe buffer (Invitrogen, USA). The Qiagen gel extraction kit was used to recover the band corresponding to each genotype of Guarani detected in Tupanvirus supernatant (wild-type and mutant) from the 2% agarose gel according to the manufacturer's instructions. Sanger sequencing analyses were then performed in an Applied Biosystems® 3130/3130xl Genetic Analyzer (Thermo Fisher, USA) using a Big Dye Terminator Cycle Sequencing Kit (Thermo Fisher, USA) according to the manufacturer's instructions. The obtained sequences were assembled with ChromasPro 1.7.7 software (Technelysium Pty Ltd, Australia).

**Passage experiments and inhibition assays.** To investigate whether mutant Guarani was able to maintain the deletion mutation during several passages with Tupanvirus, *A. castellanii* cells at a concentration of $5 \times 10^5$ cell/ml in 10 ml PYG medium were inoculated with Tupanvirus Deep Ocean and Guarani wild-type genotype at MOIs of 10. The coculture was incubated at 30 °C until complete lysis of amoebas was observed. To perform serial flask passaging, every 48 h p.i., 100 μl of the virus–virophage supernatant collected from this primo-coculture was used to infect fresh *A. castellanii* cells seeded at a $5 \times 10^5$ cells/ml density in 10 ml of PYG medium. The subculture was repeated 4 times. The same experiment was repeated by adding fresh Tupanvirus at each passage. After each passage, including the primo culture, 200 μl aliquots of the coculture were collected from the supernatant for PCR targeting the deletion site in Guarani.

To evaluate whether mutant Guarani was able to inhibit the production of Tupanvirus infectious particles during a passage experiment, *A. castellanii* cells at a $5 \times 10^5$ cells/ml density in 10 ml PYG medium were simultaneously coinfected with Tupanvirus Deep Ocean at an MOI of 10 and 100 μl of Guarani mixture containing both wild-type and mutant genotypes (according to PCR). The Guarani used here was isolated from the second sub-coculture with Tupanvirus where the mutation was clearly observed. The serial passaging experiment was performed as described above, but in this experiment, coinfected cells were observed daily and each supernatant was used to perform a new round of subculture after complete lysis of amoebas. In absence of total cell lysis, supernatants were collected at 5 days p.i. The same experiment was repeated by adding fresh Tupanvirus at each passage. After each passage, the titer of infectious particles was quantified by end-point dilution during supernatant collection and after heating at 55 °C for 30 min.

To quantify the replication of Guarani during the ten-passage experiment with Tupanvirus. At each passage, 200 aliquots of the coculture were collected for real-time PCR targeting the MCP of Guarani (Table 1), at the time of supernatant inoculation and amoebae lysis or 5 days p.i. in absence of total cell lysis. The results were analyzed by the delta Ct method between these times as described above. The same procedure was repeated by adding fresh Tupanvirus at each passage.

To study the effect of mutant Guarani on the DNA replication and virion production of Tupanvirus during a one-step growth curve, cells were coinfected with Tupanvirus Deep Ocean and mixture of wild-type and mutant Guarani at MOIs of 10. The Guarani used in this experiment was isolated and then purified from the second sub-coculture with Tupanvirus. The purification was performed as described previously[14]. To assess the effect on DNA replication, a 200-μl aliquot of the coculture was collected after 0 and 48 h p.i. and submitted to DNA extraction and then to real-time PCR that targets the MCP encoding gene in the Tupanvirus Deep Ocean (Table 1). To evaluate the effect on the production of Tupanvirus virions, a 1 ml aliquot was collected at times 0, 6, 12, 16, 24, 32, 48, and 72 h p.i. The virus supernatant was frozen and thawed three times to release the virions, and the titer was then determined at each time point by end-point dilution.

To perform the dose–response to virophage, the same procedure was repeated using different MOIs of the giant virus (Tupanvirus or APMV) and the virophage (Guarani or Zamilon). The supernatant was then collected for end-point dilution after complete lysis of cells or 5 days p.i. in the absence of cell lysis.

**Virus titration.** Virus titration was performed using end-point dilution in 96-well plates and calculated by the Reed and Muench method as previously described[14,55]. For the passage experiment and the one-step growth curve, the viral supernatants were serially diluted from $10^{-1}$ to $10^{-10}$ in 100 μl of PAS, and were then added to each well. For the dose–response experiment, the viral samples were serially diluted from $5^{-1}$ to $5^{-14}$ in 100 μl of PAS. To inhibit the virophage and avoid interference with giant virus multiplication during the end-point dilution method, each virus supernatant was heated for 30 min at 55 °C. This treatment allowed us to inactivate the virophage without decreasing in the virus titer.

**Transmission electron microscopy (TEM).** TEM experiments were performed as previously described by Mougari et al.[14]

**Ribosomal RNA shutdown and virophage replications assays.** To investigate whether the cytotoxic profile of Tupanvirus associated with ribosomal RNA (rRNA) shutdown was able to prevent virophage replication with the virus and the capacity of mutant Guarani to inhibit this phenotype, 1 million *A. castellanii* cells were infected with Tupanvirus Deep Ocean or Moumouvirus at MOIs of 100 and each virophage at MOIs of 10. The evaluation of the rRNA shutdown for each condition was conducted as previously described by Abrahao et al[16]. Briefly, one

milliliter of each coculture supernatant was collected at time 9 h p.i. and then subjected to total RNA extraction (Qiagen RNA extraction Kit, Hilden, Germany). The RNAs extracted from the samples were normalized, and then electrophoresed and visualized on an agarose 1% gel using SYBR safe buffer (Invitrogen, USA). The DNA replication of virophages was quantified as detailed above.

**Statistics and reproducibility.** Experiments were performed in biological triplicate. Mean assay from three independent biological experiments was presented in each figure, in which error bars represented standard deviation. A number n suggested biological replications. Intermediate values were used in the calculations. Unpaired two-tailed Student's $t$ test was used for comparison of two experimental groups. For all analyses, $p < 0.05$ are considered significant.

**Reporting summary.** Further information on research design is available in the Nature Research Reporting Summary linked to this article.

## Data availability
The genome of Guarani wild-type is available in the EMBL-EBI database under accession number LS999520. The collagen-like gene sequence of mutant Guarani has been deposited in the GenBank database under the accession number MT179725. The complete genome of mutant Guarani is available in Supplementary Data 2. The source data underlying plots shown in figures are provided in Supplementary Data 4. The authors declare that all other data supporting the findings of this study are available within the paper and its supplementary information files.

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

## Acknowledgements

This work was supported by a grant from the French State managed by the National Research Agency under the "Investissements d'avenir" (Investments for the Future) program with the reference ANR-10-IAHU-03 (Méditerranée Infection) and by Région Provence-Alpes-Côte- d'Azur and European funding FEDER PRIMMI. Said Mougari was supported by the Foundation Méditerranée Infection scholarship. We are grateful for Meriem Bekliz for her help and assistance at the beginning of this project.

## Author contributions

S.M. conceived the study, designed and performed the experiments, collected, analyzed and interpreted the data, generated the figures, drafted the text and wrote the manuscript. N.C. and S.M. analyzed genomic data. D.S-B. performed experiments. F.D.P. and S.M. performed electronic microscopy experiments. P.C. analyzed genomic data, reviewed the manuscript and contributed to the final version. B.L.S. and J.A. conceived the study, designed the experiments and wrote the manuscript.

## Competing interests

The authors declare no competing interests.
