## [Peer Review File · Communications Biology]

Reviewers' comments:

Reviewer #1 (Remarks to the Author):

Brief summary of the manuscript:

In the paper 'First evidence of host range expansion in virophages and its potential impact on giant viruses and host cells' Mougari S et al. isolate a new Guarani mutant able to infect the giant viruses Tupanvirus Deep Ocean (TDO) and Tupanvirus Soda Lake (TSL) which infect amoeba. Interestingly, when amoeba is coinfecting with the Guarani mutant and the TDO, Guarani mutant impairs the TDO particles production increasing amoeba survival.

Overall impression of the work:

The work presented here by Mougari S et al. is interesting for the scientific community in general. The major finding of this work is that the loss of only 81 nucleotides in a gene (27 amino acid) allows the virophage Guarani to infect another giant virus avoiding its propagation.

The paper is well written and easy to follow, however, to be totally convincing some changes should be introduced in the manuscript with further controls and additional experiments.

Specific comments, with recommendations for addressing each comment:

Even if I understand and share the difficulty of selecting the final figures for a manuscript, in the present article, some key-figures are missing. I would like to remark that this study comprises 3 main protagonists: the host cell *A. castellanii*, the virophage Guarani and the giant virus TDO or TSL having, all of them, the same importance. Therefore, each of the players should be monitored in parallel to show, jointly, the viral (giant virus and virophage) and cellular propagation/inhibition. In addition, due to the complexity of the system, negative controls are also necessary.

Major points:

1. In the introduction, authors spend some time explaining previous findings (e.g. MIMIVIRE) that are never mentioned after in the manuscript but there is no mention at all of the virophage Guarani and just a sentence on Tupanviruses (main protagonists of the study). Please, add more information about them.

2. Throughout the entire manuscript, the authors remark that the mutation found in Guarani is produced in only one passage during co-infection with both Tupanvirus (1st time mentioned on line 96). In addition, they defend that this mutation is spontaneously produced (line 298) in the virophage to allow it to infect a new giant virus ('host range expansion'). At line 99 as well as in the discussion, authors describe the possibility that this mutant was already in the original stock although they cannot detect it.

The latter is the most credible explanation and, therefore it should be taken much more in consideration in the manuscript for different reasons:

- First, in other phages/viruses (especially dsDNA virus) more than one passage is usually necessary to see the emergence of a spontaneous mutation which allow the infection of another host.
- Second, in lines 305-307 of the discussion (although authors do not show the data) it is written that, when a pure mutant Guarani is used to infect mimivirus of group A, B and C, the mutant shows the same replication than the wild-type (WT) Guarani. This strongly indicates that this mutant could also propagate together with the WT (although, probably WT wins the battle in the mixture) during the infection with mimivirus used to generate the first Guarani stock. However, when the giant virus used is Tupanvirus, the mutant is propagated more efficiently than the WT and, consequently, authors are enriching for the mutant.
- Third, in the Figure 10 panel B, authors show the mutant Guarani in the Guarani native population

which indicates that they also consider this hypothesis as the most probable.

Due to these reasons, all the manuscript (including title) should be changed to present this hypothesis as the most probable one, and use the discussion to show the possibility of a 'spontaneously' mutation and 'host range expansion'.

3. In this paper, all the experiments performed in order to characterize the mutant Guarani and its specific behavior have been done with a mixture of WT and mutant Guarani (Materials and Methods line 513 stating that authors took the passage 2 and figure 4a showing that at passage 2 both WT and mutant are present). This fact makes it impossible to discard the WT Guarani influence in the observed results. Therefore, experiments should be repeated with a pure mutant Guarani. Otherwise, in the text all mention of mutant Guarani should be defined as 'mixture of WT and mutant'.

4. In the Figure 1, it would be interesting to follow the giant virus replication by qPCR but not only the Tupanviruses but also APMV and Megavirus to exclude presence of mimivirus replication coming from the virophage stock. Although authors checked the APMV and Megavirus absence (lines 442 and 443), the data is not shown and it was not done with qPCR and at 24 hours postinfection which would be more sensible. As mentioned above, authors should also check and show the data for the presence of the giant virus along with the virophage growth (e.g. figure 1, figure 2, figure 4 etc...) and even the host propagation.

5. If the qPCR primers used for Guarani and Sputnik are the same, how can the authors be sure that they are detecting Guarani instead of Sputnik? Could the authors find another region/primers to probe that the virophage is Guarani and not Sputnik (maybe ORF18 or 20)? In the same way, the authors should do a specific qPCR in their virophage stocks to discard cross-contamination with the other virophages used in the study.

6. In materials and methods, line 451, the authors say that they use the amoeba infected only with Tupanvirus as negative control, but they do not show the qPCR data of this negative control. All the primers used for major capsid protein qPCR of Guarani, Sputnik and Zamilon can partially hybridize with TDO and TSL genome which makes it necessary to have the qPCR control results in the paper to exclude a qPCR product coming from the giant virus or from the host instead of the virophage that they are targeting.

7. How different are Sputnik and Guarani in sequence? WT (published in May 2019, reference #24) and mutant Guarani sequences are not available although authors say they have been fully sequenced. These sequences are essential for this manuscript. In addition, the mutant Guarani should be fully sequenced (again) when the authors have a pure population of the mutant, so after serial passages and not only from the mixture of WT and mutant virophages.

8. Paragraph from line 104 to 116 and Figure 2C are not conclusive, which is the Guarani preparation used here? Is it the mixture of the WT and the mutant Guarani? If it is a mixture, WT is still present and it is known that it can replicate in APMV, Moumovirus and Courdo 11. The authors discuss this point in line 305, and they say this experiment was also done with an "almost pure mutant" coming from passage 5 in Figure 4a, but they do not show this data. They should change graph in Figure 2c for one done with the "almost pure mutant", ideally with a pure mutant Guarani stock.

9. Gel B in Figure 3 (left panel) is overloaded, if there is another (slight) band coming from the mutant it is not possible to see it. This figure would be much more conclusive if both PCR products were load on the same gel with care on the quantity loaded (same quantity, same exposure). Please, show in supplementary data the sequences of the products. (See also comment #8 in minor points)

10. Figure 4a tries to show the propagation of the mutant Guarani. However, this experiment should be repeated adding fresh giant virus (with the same MOI) at every passage, which will allow the propagation and expansion of the mutant Guarani and will provide information on the stability of the mutant. Otherwise, in absence of Tupanvirus replication, Guarani replication is hindered and mutant Guarani will be lost and diluted in every passage, as it is shown in this figure. From the assumption of the authors, the WT Guarani should not replicate as it does not contain the deletion allowing for the host range expansion. As the results are shown in Figure 4 do not allow to demonstrate either the non-propagation of WT or the propagation of the mutant Guarani as authors say in lines 309-310 because both viruses are losing in every step. In addition, authors say that they wait until lysis in this experiment; does it mean that every supernatant were taken at different hours post-infection, as we can expect a delay in the cell lysis due to the absence of giant virus? If yes, please indicate the timings. If in all cases lysis was detected, it means that there is still giant virus present which would not explain the loss of the mutant Guarani. In addition, from the results on the Tupanvirus infectivity assay (figure 4c) it is shown that the giant virus is still present at passage 5 (1.10^3 TCID₅₀/ml) whereas the mutant virophage is almost lost (figure 4a). How do the authors explain the further decrease of the giant virus if there is so little competition from the virophage? One more time, both viruses together with the amoeba should be monitored in parallel to make conclusions. Was the control (Guarani + APMV) of the Figure 4b taken at the same time post infection as in the Figure 4a? If Guarani also impaired the infectivity of APMV (line 159), why is it not decreasing also in the Figure 4b? The authors do not discuss this.

11. Figure 4 panels a and b is the key image of the article. However, the presentation of the data raises concerns. Both gels share the exact same ladder (although in the original gel they have 2 different ones) and the gel from figure 4b has an extremely poor resolution compared to all the other gels presented in the article (in the original gel is overloaded). Another figure should be provided showing all the data on the same gel (including the ladder and with the same amount of PCR product) to allow clear comparison (as mentioned in the point #10 of the review).

12. Figure 4c and 5b do not show error bars. It would be also relevant if authors could show the quantification of the mutant virophage by (specific) qPCR at the same points to confirm that the increase of the mutant Guarini is responsible for the stop or decrease of the Tupanvirus propagation.

13. In figure 5b, please set the y-axis to 1. Can the authors discuss the slight increase observed in the Tupanvirus growth curve from H16 to H72? Can the authors discuss as well what they mean with the sentence: the inhibition was "significantly far higher than that of the wild-type toward APMV" (line 195) as this inhibition was estimated to be of 90% after 16 hpi.

14. In lines 188 and 189, the authors say that 'the early DNA replication of giant virus may allow them to anticipate virophage parasitism and produce some genomic copies'. This sentence tries to explain why Guarani does not affect the Tupanvirus DNA replication but the timing of giant DNA replication and its protection against virophages do not have to be related, simply, the virophage inhibits giant virus propagation in another point of the cycle or through another mechanism. So, please, remove this sentence. Also, can the authors specify which type of infectious virions (giant virus (APMV or Tupanvirus) or virophage) they were able to detect (line 196)?

15. Panel D of figure 5 should be removed. It does not give any information, if panel C already shows that Guarani is always found in amoeba when there is no Tupanvirus associated, hence it does not make sense to count the number of Tupanvirus particles present per cell. The fact that only the amoebas, in which Guarani was detected, were taken into account to analyze the presence of Tupanvirus must be remarked in the text of figure 5 panel D and also in more detail in the text. Otherwise, this result is confusing and opposite to panel b and the text in line 226.

16. Lines 217 and 218, 'In this study, we clearly observed that these small viruses obtain essential elements from the factories of their host viruses to propagate'. This manuscript does not analyze that, and this cannot be observed using only electron microscopy. So, please, just suggest this hypothesis based on the co-localization of the Guarani and the viral factory, but do not jump to conclusions.

17. In figure 9a, the gel size standard is missing.

18. Figure 10 panel D, the authors say that there is no Tupanvirus replication, but this is not shown or commented in the manuscript. In addition, the lack of virophage replication in this condition could have others reason than the shutdown of the host rRNA, and more, taking into account that virophage replication most likely depends on the giant virus.

What is a low to moderate MOI? Indicate this in the figure.

Minor points:

1. Figure 2 (the schematic summary) is the Figure 10. All Figures are wrongly listed

2. Line 44 --- remove the parenthesis (a protist)

3. Line 53 ---remove the parenthesis (e. g., Zamilon). Authors are already, and only, talking about Zamilon in this paragraph

4. Line 85, are there still amoeba cells alive at 48 hours post-infection? Why between 24 hours and 48 hours Spuntik and Zamilon do not continue to replicate (figure 1)?

5. Line 92, the authors say that the obtained supernatants containing only Guarani particles but, they do not explain how they discard the giant virus in this preparation (neither in materials and methods). Nothing is said concerning the MOI used.

6. Line 122, Guarani genome consists of a circular dsDNA. Circularity was not experimentally proved. Remove circular

7. Line 130, Table S1 does not exist

8. In Figure 3b (right panel), the authors show a PCR product coming from Guarani produced with Tupanvirus, but they do not indicate which Tupanvirus they are using here. It will be helpful to see the PCR product coming from both Tupanviruses used as they say, in lines 130 and 131, that this mutation appears during the coinfection with both Tupanviruses

9. Line 137, the authors say that the sequencing of the PCR product allow them to localize the mutation in the ORF8 of the virophage, but that is not possible, because the target of the PCR is already the ORF8 so, they cannot extract this information from this PCR. They can only conclude that the deletion is present. Please, remove this sentence

10. Figure 4c, what is the MOI corresponding to 100 μ L of the mixture of WT and mutant Guarani?

11. Line 168, add the word giant in the sentence 'viable virus particles'

12. Line 174, 'mutation increased its virulence', but, comparing with what? In this experiment there is no comparison, so the mutation cannot increase its virulence

13. Line 191, 'that mutant Guarani induces a severe decrease in the production of Tupanvirus virions'. Change production by infectivity (end-point dilution assay measure the infectious particles no the total number of particles)

14. Figure 5f-h shows always the same cell. Could be possible to see more cells instead of two different zooms?

15. Line 209 and text figure 6, indicate the times post-infection. Can the authors discussed the difference of replication rates observed with the WT Guarani on APMV (reference 24 figure 1) and the mutant Guarani on TDO?

16. Line 219, add giant before 'virus'

17. Line 244, add reference to Figure 7c

18. The authors say that a subpopulation of Sputnik also showed the same deletion as the mutant Guarani (line 344). Unfortunately, the authors never used the mixture of WT and mutant Sputnik as a

comparative virophage in their study. They preferred to use Zamilon, a virophage known to have no effect in giant viruses replication so far. So, next question arises, how different are Guarani and Sputnik during Tupanvirus infection? Comparative study would be very interesting

19. Line 447, how was the quantification of the TDO and TSL done?

20. Line 453, were the 200 μ L aliquot taken from the supernatant or cells were taken and lysed to do the qPCR?

21. Line 515, add that, before doing the end-point dilution, the viruses were heating at 55°C, 30 min

22. Line 520, how was the Guarani purified?

23. Figure 9, add the MOI of virophage in the figure text

24. The panels in the text of figure 10 are not well indicated

25. Panel c of figure 10, authors are not using a pure mutant Guarani if not a mixture of WT and mutant, in this panel only the mutant is shown

26. In future works, the authors should try to use the same infection condition (same MOIs in the different experiments) at least that the use of different conditions is justified to arrive at a conclusion. In this paper, different experiment are not comparable and make the understanding difficult.

Reviewer #2 (Remarks to the Author):

As we know, defense systems in bacteria have been identified and demonstrated rather clear, however, it is still unclear in viruses, we knew that virus are specific for cognate host. This manuscript presented by Mougari S, et al., which reported virophage could abolish giant virus production and rescue the host cell population from virus lysis. The story presented seems well rounded, particularly; it should be interested in this field. Some issues will be addressed as following.

Major concerns: The manuscript describes that the deletion of ORF8 involved in integrating Guarani into the genome of Tupanvirus to allow the virophage to replicate. Here, there is no solid data to support this, if the authors could provide supportive data for this, the manuscript would be better.

Minor remarks.

1/ Given how the study is, the introduction would be concise and focused for the reader to gain a basic understanding of host range expansion in a virophage.

2/ lines142-144 could be modified the sentence for clearance.

3/ lines177-180 Could be consolidated into more single statement.

4/ Other statements could be clear and short, for example, lines300-323.

5/ Could be given the tittle of each figures.

Point-by-point response to Reviewers

Point-by-point response to Reviewer #1 comments

Reviewer: Brief summary of the manuscript:

In the paper 'First evidence of host range expansion in virophages and its potential impact on giant viruses and host cells' Mougari S et al. isolate a new Guarani mutant able to infect the giant viruses Tupanvirus Deep Ocean (TDO) and Tupanvirus Soda Lake (TSL) which infect amoeba. Interestingly, when amoeba is coinfecting with the Guarani mutant and the TDO, Guarani mutant impairs the TDO particles production increasing amoeba survival.

Overall impression of the work:

The work presented here by Mougari S et al. is interesting for the scientific community in general. The major finding of this work is that the loss of only 81 nucleotides in a gene (27 amino acid) allows the virophage Guarani to infect another giant virus avoiding its propagation.

The paper is well written and easy to follow, however, to be totally convincing some changes should be introduced in the manuscript with further controls and additional experiments.

ANSWER: *Dear peer reviewer, thank you for your comments, we tried to follow all your recommendations. We have made additional experiments, substantial changes to the text and have redone most of the figures to improve the quality of the manuscript and result presentation.*

Reviewer: Specific comments, with recommendations for addressing each comment:

Even if I understand and share the difficulty of selecting the final figures for a manuscript, in the present article, some key-figures are missing. I would like to remark that this study comprises 3 main protagonists: the host cell *A. castellanii*, the virophage Guarani and the giant virus TDO or TSL having, all of them, the same importance. Therefore, each of the players should be monitored in parallel to show, jointly, the viral (giant virus and virophage) and cellular propagation/inhibition. In addition, due to the complexity of the system, negative controls are also necessary.

ANSWER: *Please see the answers below.*

Major points:

Reviewer: 1. In the introduction, authors spend some time explaining previous findings (e.g. MIMIVIRE) that are never mentioned after in the manuscript but there is no mention at all of the virophage Guarani and just a sentence on Tupanviruses (main protagonists of the study). Please, add more information about them.

ANSWER: *According to the reviewer suggestion, we have shortened the part of the introduction discussing the MIMIVIRE system, and more information was added regarding the virophage Guarani and the Tupanviruses.*

Reviewer: 2. Throughout the entire manuscript, the authors remark that the mutation found in Guarani is produced in only one passage during co-infection with both Tupanvirus (1st time mentioned on line 96). In addition, they defend that this mutation is spontaneously produced (line 298) in the virophage to allow it to infect a new giant virus ('host range expansion'). At line 99 as well as in the discussion, authors describe the possibility that this mutant was already in the original stock although they cannot detect it.

The latter is the most credible explanation and, therefore it should be taken much more in consideration in the manuscript for different reasons:

- First, in other phages/viruses (especially dsDNA virus) more than one passage is usually necessary to see the emergence of a spontaneous mutation which allow the infection of another host.
- Second, in lines 305-307 of the discussion (although authors do not show the data) it is written that, when a pure mutant Guarani is used to infect mimivirus of group A, B and C, the mutant shows the same replication than the wild-type (WT) Guarani. This strongly indicates that this mutant could also propagate together with the WT (although, probably WT wins the battle in the mixture) during the infection with mimivirus used to generate the first Guarani stock. However, when the giant virus used is Tupanvirus, the mutant is propagated more efficiently than the WT and, consequently, authors are enriching for the mutant.
- Third, in the Figure 10 panel B, authors show the mutant Guarani in the Guarani native population which indicates that they also consider this hypothesis as the most probable.

Due to these reasons, all the manuscript (including title) should be changed to present this hypothesis as the most probable one, and use the discussion to show the possibility of a 'spontaneously' mutation and 'host range expansion'.

ANSWER: *We totally agree with this comment. As explained by the reviewer, we consider the scenario under which, the mutant was already present in our virophage stock before the challenge with Tupanvirus, as the most probable one (see Line 104 please). In any case we meant that Guarani mutant emerged (biologically) after a first passage in cells co-infected with Tupanvirus. Apologizes if this was not clear enough. Indeed, if we could not see wild-type Guarani replication in such first passage with Tupanvirus, there would not be possible the generation of mutants virophages, because the generation of mutants requires virophage replication (genome, proteins, assembly, morphogenesis, etc). Maybe such mutant came from cells coinfecting by Mimivirus and Guarani wild type, in which were used for the production of Guarani stocks.*

As stated in the manuscript Line 33, the expansion of host-range requires mutation. But this mutation most likely takes place during pathogen replication at the original host, raising a

mixed population containing WT and mutants. Then, when mutants get contact with other hosts, there is the possibility of cross-species infection.

The term “spontaneous” in this manuscript was not used to designate the spontaneous occurrence of host range expanding mutation in Guarani genome once this virophage was challenged with Tupanvirus. This term was used to indicate that this mutation has occurred naturally (during propagation in nature or production at the lab) and was not induced by genome editing like it was the case for other studies cited in the manuscript (See Line 112 please). For example, replacing the host-determinant region of long tail fiber gene in the T4-like phage, WG01, with that of another T4-like phage, named QL01, enabled the recombinant phage to infect both WG01 and QL01 hosts (PMID: 28939606).

Therefore, the entire manuscript, including the title and the abstract, has been modified to present more clearly the adaptation of Guarani to Tupanvirus as, most likely, the result of selection of a mutant resulting in host range expansion of the virophage. To avoid any confusion, the expression “host range expansion” was replaced by cross-species infection or “selection of a mutant resulting in host range expansion” throughout the manuscript.

Reviewer: 2. (This strongly indicates that this mutant could also propagate together with the WT (although, probably WT wins the battle in the mixture) during the infection with mimivirus used to generate the first Guarani stock).

ANSWER: *This was added in the revised manuscript Line 369-thank you.*

Reviewer: 3. In this paper, all the experiments performed in order to characterize the mutant Guarani and its specific behavior have been done with a mixture of WT and mutant Guarani (Materials and Methods line 513 stating that authors took the passage 2 and figure 4a showing that at passage 2 both WT and mutant are present). This fact makes it impossible to discard the WT Guarani influence in the observed results. Therefore, experiments should be repeated with a pure mutant Guarani. Otherwise, in the text all mention of mutant Guarani should be defined as ‘mixture of WT and mutant’.

ANSWER: *We took subculture 2 and not passage 2 as corrected in the revised version (Line 167). But we agree and according to this comment, all mention of mutant Guarani was replaced by “mixture of wild-type and mutant” or “mixture containing mutant Guarani” in the revised version of the manuscript. Although we agree with the reviewer suggestion, repeating all the experiments of this study with a pure mutant requires significant time and is not possible to set up for us. In addition, even in Guarani isolated from passage 5 (or further) with Tupanvirus where PCR was negative for the wild-type, we could not discard the possibility of presence of the wild-type at a slight concentration (as we do not discard the possibility of presence of mutant in the original stock even not detected). However, we believe that the results we obtained using the mixture reflect the behavior of the mutant as the WT was neither able to replicate with Tupanvirus nor to cause any inhibition to this giant virus as mentioned in the manuscript. That said, two questions need to be addressed in further studies, the viability*

of the WT present after first passages with Tupanvirus and the capacity of the mutant to replicate and cause this inhibition profile alone in absence of the WT. This was mentioned as perspectives at the end of the revised manuscript. Thank you.

Reviewer: 4. In the Figure 1, it would be interesting to follow the giant virus replication by qPCR but not only the Tupanviruses but also APMV and Megavirus to exclude presence of mimivirus replication coming from the virophage stock. Although authors checked the APMV and Megavirus absence (lines 442 and 443), the data is not shown and it was not done with qPCR and at 24 hours postinfection which would be more sensible. As mentioned above, authors should also check and show the data for the presence of the giant virus along with the virophage growth (e.g. figure 1, figure 2, figure 4 etc...) and even the host propagation.

ANSWER: *As suggested, all the DNAs extracted from the cocultures of experiments in Figure 1 and Figures 2a and 2b (first version of the manuscript) at 0h, 24h and 48h were checked for the replication of APMV (in Guarani and Sputnik cocultures) and Megavirus Courdo 11 (in Zamilon cocultures). As expected, all the qPCRs were negatives (N/A: No amplification). We believe that adding these negative controls in the figure is not necessary, so, we preferred to only state this in the text of each figure. In the same manner, following the replication of Tupanvirus (or the mimiviruses in figure 2) at this step by qPCR concurrently with the virophages is not the aim of this figure. This has been done in Figure 4a (revised version) for Tupanvirus in presence of Guarani (and please note that given the giant size of these viruses, their multiplication could be observed by optic microscopy). It is the same for the host cell population growth (done in Figure6). We tried to respect a chronology in presenting the results and to keep only one information presented for each figure in order to avoid any confusion, and the major result to show for Figure 1 (Fig1 and Fig 2a-2b in the first version) was the host specificity of Tupanviruses to Guarani. The interactions between these viruses was characterized in the other figures of the manuscript.*

Reviewer: 5. If the qPCR primers used for Guarani and Sputnik are the same, how can the authors be sure that they are detecting Guarani instead of Sputnik? Could the authors find another region/primers to probe that the virophage is Guarani and not Sputnik (maybe ORF18 or 20)? In the same way, the authors should do a specific qPCR in their virophage stocks to discard cross-contamination with the other virophages used in the study.

ANSWER: *The virophage Guarani was sequenced following its first production, immediately before the beginning of the experiments for this study as it was isolated recently. The sequencing confirmed that this virophage was a pure new Sputnik-like virus. All virophages used in this study were produced and purified separately in time (more than three months between each production). Moreover, the coinfections with Tupanviruses were first performed with only one virophage at a time. This experiment was first done for Sputnik and Zamilon (separately) and then few months later for Guarani when this virophage was isolated (and immediately after Guarani genome sequencing). Afterward, the results of coinfections were reproduced using Tupanviruses and the three virophages in the same experiment and in*

triplicate. These experiments have been performed very delicately, as detailed, making almost impossible to have cross contaminations between virophage stocks (See paragraph in Line 529 – 534 please). In addition, while the same primers were used to quantify the replication of Sputnik and Guarani, the genome of the latter was sequenced again before and after passage in Tupanvirus and no Sputnik contamination was found in both conditions. Specific PCR systems to discard cross-contamination were also performed after the production of each virophage. The presence of Zamilon in Sputnik stock was negative and vice-versa. We also designed a PCR system targeting ORF19 (Table 1), which is specific to Guarani and confirmed, as expected, that we were detecting Guarani. This system was also used to discard the presence of Guarani in Sputnik and Zamilon stocks according to the reviewer suggestion (see paragraph in Line 513 – 517 please). Unfortunately, however, given the high similarity between Guarani and Sputnik, it is not possible to design a specific PCR system that detect Sputnik and not Guarani. In this case, genome sequencing was the only procedure used to confirm the absence of Sputnik in Guarani stock. As precised in the revised manuscript (paragraph in Line 542 – 547), the new qPCR specific to Guarani and not Sputnik was used to quantify the replication of this virophage in primo-coculture with APMV, Tupanviruses and then during subcultures with Tupanviruses. These results confirmed our previous results regarding the host specificity of Tupanvirus to Guarani and the capacity of the latter to propagate in subcultures with this virus. However, we believe that given that cross contamination between Sputnik and Guarani was discarded, using the same primers to quantify the replication of these virophages will allow readers to compare their replication more precisely. In our study, Sputnik was used as control for replication, so we prefer to present in Figure 1 (revised version) the results obtained using the same primers for these two Sputnik-like virophages. All these points have been added in the materials and methods part of the revised manuscript.

Reviewer: 6. In materials and methods, line 451, the authors say that they use the amoeba infected only with Tupanvirus as negative control, but they do not show the qPCR data of this negative control. All the primers used for major capsid protein qPCR of Guarani, Sputnik and Zamilon can partially hybridize with TDO and TSL genome which makes it necessary to have the qPCR control results in the paper to exclude a qPCR product coming from the giant virus or from the host instead of the virophage that they are targeting.

ANSWER: *We agree with this suggestion. Virophage and mimivirus genomes usually show sequence similarities. And we always check this parameter during primer design and PCR tests. As stated in figure texts of the revised manuscript, all the virophage PCR systems targeting the amoebas incubated with each giant virus used in this study, in absence of virophages, were negatives.*

Reviewer: 7. How different are Sputnik and Guarani in sequence? WT (published in May 2019, reference #24) and mutant Guarani sequences are not available although authors say they have been fully sequenced. These sequences are essential for this manuscript. In addition, the mutant Guarani should be fully sequenced (again) when the authors have a pure population of the mutant, so after serial passages and not only from the mixture of WT and mutant virophages.

ANSWER: *The comparison between Guarani and Sputnik genomes has been detailed in our previous paper (PMID: 31130943). The genome of WT Guarani is now released under the accession number (GenBank: LS999520.1). Genomes of Guarani WT and mutant are now attached in supplementary data (Table S1) as suggested (submitting them to GenBank will take significantly more time). On the other hand, the Guarani produced after several passages with Tupanvirus was already sequenced and showed the mutant sequence described in this study as stated in the revised manuscript (Line 306). However, we believe that it is not possible to call this strain pure as it may contain a wild-type background. Probably, the only way to discard this contamination is to perform several rounds of end-point dilution coupled with PCR screening of each clone. Unfortunately, these experiments require significant time and work and even that this, probably, could not be enough to purify the mutant.*

Reviewer: 8. Paragraph from line 104 to 116 and Figure 2C are not conclusive, which is the Guarani preparation used here? Is it the mixture of the WT and the mutant Guarani? If it is a mixture, WT is still present and it is known that it can replicate in APMV, Moumavirus and Courdo 11. The authors discuss this point in line 305, and they say this experiment was also done with an “almost pure mutant” coming from passage 5 in Figure 4a, but they do not show this data. They should change graph in Figure 2c for one done with the “almost pure mutant”, ideally with a pure mutant Guarani stock.

ANSWER: *We agree with this suggestion. So, the results of the experiment done with supernatant coming from passage 5 with Tupanvirus was presented instead of the previous figure. The results have been placed at the end of the result part of this manuscript because they were performed after the passaging experiment presented in Figure 3 (revised version). Only the results of Guarani isolated from the Tupanvirus Deep Ocean were presented because this virophage was the only one to be well characterized and enriched by passaging. The behavior of Guarani mutant coming from Tupanvirus Soda Lake should be characterized in further studies as precised in the revised manuscript. (please see part: Analysis of mutant Guarani fitness with the prototype isolates of the family Mimiviridae, in the results)*

Reviewer: 9. Gel B in Figure 3 (left panel) is overloaded, if there is another (slight) band coming from the mutant it is not possible to see it. This figure would be much more conclusive if both PCR products were loaded on the same gel with care on the quantity loaded (same quantity, same exposure). Please, show in supplementary data the sequences of the products. (See also comment #8 in minor points).

ANSWER: *We agree with this suggestion. The experiment was repeated. Both PCR products were loaded in the same gel with same quantity and the same exposure. The migration time has been extended (45 min instead 30 min) to allow a better separation of the mutant and wild-type bands. The region of the gel that has the same size as the mutant band in the PCR product, coming from Guarani propagated with APMV, was extracted delicately and submitted to Sanger sequencing but, as expected, no mutant sequence was detected. The sequences of the products were shown in supplementary data (Table S1) of the revised manuscript.*

Reviewer: 10. Figure 4a tries to show the propagation of the mutant Guarani. However, this experiment should be repeated adding fresh giant virus (with the same MOI) at every passage, which will allow the propagation and expansion of the mutant Guarani and will provide information on the stability of the mutant. Otherwise, in absence of Tupanvirus replication, Guarani replication is hindered and mutant Guarani will be lost and diluted in every passage, as it is shown in this figure. From the assumption of the authors, the WT Guarani should not replicate as it does not contain the deletion allowing for the host range expansion. As the results are shown in Figure 4 do not allow to demonstrate either the non-propagation of WT or the propagation of the mutant Guarani as authors say in lines 309-310 because both viruses are losing in every step.

ANSWER: *According to this suggestion, the experiment in Figure 4 (now Fig. 3c) has been repeated by adding fresh Tupanvirus to allow the propagation of mutant Guarani. The results confirm that this repeated passages with Tupanvirus really promotes the expansion of the mutant rather than the WT, as explained in the revised version of the manuscript Line 161.*

Reviewer: 10. In addition, authors say that they wait until lysis in this experiment; does it mean that every supernatant were taken at different hours post-infection, as we can expect a delay in the cell lysis due to the absence of giant virus? If yes, please indicate the timings.

ANSWER: *As stated in the manuscript, the presence of the mutant was coupled with a delay in the time of lysis of cells. Several previous studies have demonstrated that virophages cause a delay in the cell lysis (PMID: 18690211, PMID: 27929021), including Guarani (PMID: 31130943). In our study this delay could also be explained by the interruption of giant virus production in coinfecting cells. So, as indicated in the revised version line 142, in Figure 4a (now Fig. 3a) each supernatant was taken to perform a new passage at time 48 h p.i. We waited until total lysis only for the primo culture. In this case, lysis was observed before the time 48 h p.i.*

Reviewer: 10. If in all cases lysis was detected, it means that there is still giant virus present which would not explain the loss of the mutant Guarani.

ANSWER: *Total lysis was not detected in all cases at 48h p.i. This parameter was not monitored precisely during the experiment shown in Figure 3a as it is not the aim here.*

As mentioned in this manuscript according to electronic microscopy quantification, even at MOI of 10 of the virophage, we were not able to reach a rate of infection of 100% of cells. Furthermore, for experiment in Figure 3a, passages have been started by a primo infection between the WT and Tupanvirus. Thus, the MOI of mutant Guarani here is totally unknown. This means that the system is very complex, as at each passage we can find in each coculture different populations of cells; cells infected by Tupanvirus and mutant Guarani, cells infected by Tupanvirus and the remaining wild-type Guarani and cells infected only by Tupanvirus (In addition to amoebas infected by Tupanvirus and both mutant and WT). This makes it more

difficult to predict precisely what is the contribution of each part in the general behavior of the system.

The loss of mutant could be due to several reasons. First, the fact that Guarani significantly delays cell lysis and that all supernatants were taken at the same time means that an increasing number of cells was present over passages. So, progressively the supernatant collected was enriched with more amoebas. However, we know that cells infected only by the giant virus lyse more quickly than coinfecting cells. Thus, amoebas that remain in the supernatant are mainly those containing the virophage. So, we could speculate that at each supernatant inoculation, free Tupanvirus present in the supernatant may have the opportunity to infect (and lyse) the fresh amoebas while the virophage stays trapped in its cells before their lysis. But we also know that the system contains a limited number of amoebas and that even after lysis of coinfecting cells, the virophage will not have the time to replicate as the supernatant will be taken at time 48h p.i. to perform a new passage. In this manner, this phenomenon gets worse as you go with passages causing a progressive dilution of the mutant. This point was stated in the revised version Line 157. On the other hand, the virophage replicates with high replication efficiency in coinfecting cells and inhibits Tupanvirus spread. Hence, the competition between these two phenomena probably determines the general behavior of the system and the final evolution of both viruses. The fact that adding fresh giant virus in the system allows the propagation of the virophage supports this hypothesis (by acceleration of cell lysis). In addition, during passages with APMV, the virophage does not seem to be diluted. In this case, almost all cells were lysed at 48h p.i. This observation seems to reinforce the scenario given above.

Second, the virulence of Guarani suggests that the giant virus will be diluted at each passage. Hence, at every passage, cells would be infected by a lowest MOI of the virus. This means that there is more chance that some cells could be infected only by giant virus progeny with a lower reproductive ratio, which will probably influence the fitness of reproduction of the virophage in these cells. By adding fresh giant virus, we were able to save the mutant virophage and allow its propagation. In addition, we have also shown that Guarani inhibits APMV at a lower level, and in this case the virophage was not diluted, which probably reinforces this hypothesis.

We believe that the complexity of this system and this kind of passage-experiments, in which the MOIs of each actor at each step is undetermined, makes it difficult to give a precise explanation of what we are observing. That said, passage experiment could not be the best method to study the interactions within a such complex system. For this reason, we chose in this work to study the interaction between each part of the tripartite, virophage-giant virus-amoebae, during a one-step growth curve using well determined MOIs of the giant virus and of the virophage mixture.

Reviewer: 10. In addition, from the results on the Tupanvirus infectivity assay (figure 4c) it is shown that the giant virus is still present at passage 5 ($1 \cdot 10^3$ TCID₅₀/ml) whereas the mutant virophage is almost lost (figure 4a). How do the authors explain the further decrease of the giant virus if there is so little competition from the virophage?

ANSWER: Apologies for not being precise enough because experiment in Figure 4a and 4c (now 3a and 3d) were not performed in the same conditions. As indicated in materials and methods of the old version and now in the results of the revised version, experiment in Fig 4a has been started by the primo coculture (WT + Tupanvirus) while experiment in Fig 4c has been started by the mixture at an MOI of 10. The time of incubation for each passage was also different. As indicated in the revised version, in figure 4a, passages were done every 48h p.i. In Fig 4 c, supernatants were collected after complete lysis or at up-to 5 days p.i. in absence of total lysis. This because, these two figures have not the same purpose. Experiment in Fig 3a aimed to study the stability of the mutant, and during this study we noticed the increase in the host cell survival and the decrease in the mutant band as stated in the manuscript. For this reason, we decided to repeat the experiment with the mixture containing the expanded mutant to confirm its virulence before shifting to inhibition assays in one-step growth curves with well determined conditions (as explained above). Therefore, these experiments could not be compared, and passages in Figure 4a (old version) are not the same in Fig. 4c. To avoid any confusion, we changed the term passage (P) in Fig 4a (no Fig 3a) by primo culture and subcultures (Primo and Sub, respectively).

On the other hand, PCR performed on a virophage supernatant detects both DNA from infectious and noninfectious particles. And, after the lysis, free virophage DNA could be released, which will make also such a PCR positive. This suggests that, at each passage, we were not quantifying only the virophage able to cause the inhibition of the giant virus. Furthermore, along the passages, the different proportions of these DNAs (Infectious, noninfectious and free DNA) could differ. For example, probably the slight band detected at passage 5 comes mainly from infectious particles able to cause the inhibition of Tupanvirus. This is only to show that, as explained in the manuscript, PCR is a not a good technic to quantify the infectious virophage. Second, the complete elimination of Tupanvirus means that the virophage was still present until the last passage even with low concentration, sometimes close to the limit of detection, and that this low concentrated virophage could have more evident effect on Tupanvirus at a $1 \cdot 10^3$ TCID₅₀/ml. As detailed in the revised version of the manuscript Line 189, figure 3e shows that even at passage 6 and 7 we detect replication of the virophage which support its role in the inhibition of Tupanvirus. Third, it is known from a previous study based on mathematical models that virophage could select giant virus with a lower reproductive ratio. Perhaps, this is another parameter that we have to take into account. Otherwise, once again, these experiments could not be compared.

Reviewer: 10. One more time, both viruses together with the amoeba should be monitored in parallel to make conclusions.

ANSWER: We agree, the replication of Guarani during passages was monitored to confirm its role in the inhibition of Tupanvirus infectivity. The experiment was also done by adding fresh giant virus according to the reviewer suggestion. Study host cell survival was done in Figure 6 (revised version) in parallel with the giant virus infectious particle production and virophage replication. Here we preferred to only stat their evolution during passages in presence of

mutant virophage and the consequence of what has been observed in virophage propagation (see Line 157 in the revised version please).

As described above, the major aim of this figure was to investigate the stability of the mutant during passage experiment and we believe that, now, this parameter has been verified. So, the purpose here was not to study virophage effect on host cell survival in such conditions. Moreover, showing the effect on host cell survival here will make the study done during one-step growth curve obsolete. But we believe that is more relevant to study this parameter during one-step growth curve using well determined MOIs as done in Figure 6. This also will avoid redundancy and confusion of readers with too much data.

Reviewer: 10. Was the control (Guarani + APMV) of the Figure 4b taken at the same time post infection as in the Figure 4a? If Guarani also impaired the infectivity of APMV (line 159), why is it not decreasing also in the Figure 4b? The authors do not discuss this.

ANSWER: *The control in Figure 4b was taken at 48h p.i. at each passage. At that time, a complete lysis of amoebas was observed. We agree that Guarani impairs the infectivity of APMV, but it is very low compared to the inhibition of Tupanvirus. So, probably, the level of inhibition of the giant virus here and the delay in cell lysis are not enough to cause an evident impact on virophage propagation and evolution at a such short term as explained above. Moreover, APMV and Tupanvirus are two different giant virus and probably their susceptibility to virophage could be different.*

Reviewer: 11. Figure 4 panels a and b is the key image of the article. However, the presentation of the data raises concerns. Both gels share the exact same ladder (although in the original gel they have 2 different ones) and the gel from figure 4b has an extremely poor resolution compared to all the other gels presented in the article (in the original gel is overloaded). Another figure should be provided showing all the data on the same gel (including the ladder and with the same amount of PCR product) to allow clear comparison (as mentioned in the point #10 of the review).

ANSWER: *As suggested the experiment was repeated and PCR products were loaded in the same gel (same quantity, same exposure). PCR products coming from passages performed by adding fresh giant virus, at the same MOI and at each passage, were also loaded in the same gel (same quantity, same exposure). The images of different conditions were extracted from the original gel and presented separately in Figure 3 (revised version) to respect the figure design, but the original gel was attached with revised version.*

Reviewer: 12. Figure 4c and 5b do not show error bars.

ANSWER: *Error bars have been added for Figure 4c and 5b- Thank you.*

Reviewer: 12. It would be also relevant if authors could show the quantification of the mutant virophage by (specific) qPCR at the same points to confirm that the increase of the mutant Guarani is responsible for the stop or decrease of the Tupanvirus propagation.

ANSWER: *We agree that it is interesting to target the mutant with a specific qPCR to follow its replication and link it to the effect on Tupanvirus. However, we have previously tried this several times and, unfortunately, it is not possible because of the repetitive consensus that characterizes the collagen-like protein. As detailed in the manuscript, the deletion in the mutant targets a region highly infiltrated by collagen repeats, so the only region in this gene on which the primers can bind is around the five repetitive sequences as shown in Figure 3 (Figure 2 revised version).*

Reviewer: 13. In figure 5b, please set the y-axis to 1.

ANSWER: *This has been done- than you.*

Reviewer: 13. Can the authors discuss the slight increase observed in the Tupanvirus growth curve from H16 to H72?

ANSWER: *The slight increase of Tupanvirus titer observed from H12 to H72 is probably due to the increasing morphogenesis of Tupanvirus virions in cells infected only by the giant virus, as mentioned in the revised manuscript Line 215.*

Reviewer: 13. Can the authors discuss as well what they mean with the sentence: the inhibition was “significantly far higher than that of the wild-type toward APMV” (line 195) as this inhibition was estimated to be of 90% after 16 hpi.

ANSWER: *The inhibition of Guarani WT to APMV was investigated in our previous study (PMID: 31130943). In that experiment, the supernatant was collected at 48h p.i. after the lysis of amoebas. In that condition, the inhibition of APMV infectivity was estimated to be of 90%, which means a 10-fold reduction after the lysis (as stated in the revised version Line 213). But, Figure 5b shows that Guarani induced a 1000-fold reduction of Tupanvirus titer compared to the control in absence of virophage (which is far higher than WT with APMV).*

Reviewer: 14. In lines 188 and 189, the authors say that ‘the early DNA replication of giant virus may allow them to anticipate virophage parasitism and produce some genomic copies’. This sentence tries to explain why Guarani does not affect the Tupanvirus DNA replication but the timing of giant DNA replication and its protection against virophages do not have to be related, simply, the virophage inhibits giant virus propagation in another point of the cycle or through another mechanism. So, please, remove this sentence. Also, can the authors specify which type of infectious virions (giant virus (APMV or Tupanvirus) or virophage) they were able to detect (line 196)?

ANSWER: *We agree, so, this speculative sentence was removed and replaced by the reviewer explanation. The virus detected at 48h by end point dilution was Tupanvirus as specified in the revised manuscript- thank you.*

Reviewer: 15. Panel D of figure 5 should be removed. It does not give any information, if panel C already shows that Guarani is always found in amoeba when there is no Tupanvirus associated, hence it does not make sense to count the number of Tupanvirus particles present per cell.

ANSWER: *We agree, so Figure 5d was removed as suggested.*

Reviewer: 15. The fact that only the amoebas, in which Guarani was detected, were taken into account to analyze the presence of Tupanvirus must be remarked in the text of figure 5 panel D and also in more detail in the text.

ANSWER: *This was added as suggested in the revised manuscript (Line 223) and the text of the figure-thank you.*

Reviewer: 15. Otherwise, this result is confusing and opposite to panel b and the text in line 226.

ANSWER: *We were trying to explain that the 26% of amoebas infected only by Tupanvirus are the origin of the virions produced at the end of the cycle analyzed in Figure 5b. The quantification in Fig 5D was done manually by observing electronic microscopy images, so naturally, it may not reflect the real infection rates in the system and the exact number of virions produced in each cell. In addition, transmission electronic microscopy shows only one section plane and not all the totality of the cell. That is why we had to confirm the conclusion drawn from these observations by a series of biological assays.*

Reviewer: 16. Lines 217 and 218, 'In this study, we clearly observed that these small viruses obtain essential elements from the factories of their host viruses to propagate'. This manuscript does not analyze that, and this cannot be observed using only electron microscopy. So, please, just suggest this hypothesis based on the co-localization of the Guarani and the viral factory, but do not jump to conclusions.

ANSWER: *We agree with this comment and the sentence was modified as suggested-thank you.*

Reviewer: 17. In figure 9a, the gel size standard is missing.

ANSWER: Yes, we previously presented this figure in this manner in our previous paper in which we described Tupanvirus and its cytotoxic profile (PMID: 29487281) as we interest only on the ribosomal RNA. We possess another image with the gel standard size but, unfortunately, the quality is bad and could not be presented in the manuscript.

Reviewer: 18. Figure 10 panel D, the authors say that there is no Tupanvirus replication, but this is not shown or commented in the manuscript.

ANSWER: The complete description of the cytopathic profile caused by the ribosomal shutdown induced by Tupanvirus was done in a previous study (PMID: 29487281). In that study, Jonatas et al shows that Tupanvirus at high MOI causes the shutdown but do not replicate. Here we only wanted to investigate its impact on viroplasm parasitism. According to the reviewer suggestion, this was now commented in the revised version Line 326.

Reviewer: 18. In addition, the lack of viroplasm replication in this condition could have others reason than the shutdown of the host rRNA, and more, taking into account that viroplasm replication most likely depends on the giant virus.

ANSWER: Sorry for not being precise enough. We agree that it is suggested in this study that the lack of viroplasm replication is a consequence of shutdown of the host rRNA caused by Tupanvirus. But the relationship of causality described here could be indirect. For example, Tupanvirus causes the ribosomal shutdown which will kill the host cell avoiding the giant virus propagation, but, in turn, this will avoid also the propagation of viroplasm. Otherwise, in any case we aimed to determine how this cytotoxic profile could prevent viroplasm infection. We only wanted to investigate its impact. But reviewer suggestion has been stated in the revised version Line 467 of the revised manuscript.

Reviewer: What is a low to moderate MOI? Indicate this in the figure.

ANSWER: In the case of this study; low MOI means less than 1. Moderate MOI is between 1 and 10. High MOI is more than 100. This was added in the figure text- thank you.

Minor points:

Reviewer: 1. Figure 2 (the schematic summary) is the Figure 10. All Figures are wrongly listed

ANSWER: This was corrected- thank you. We also replaced the schematic summary by another one with higher quality of design.

Reviewer: 2. Line 44 --- remove the parenthesis (a protist)

ANSWER: This was done- thank you.

Reviewer: 3. Line 53 ---remove the parenthesis (e. g., Zamilon). Authors are already, and only, talking about Zamilon in this paragraph

ANSWER: *This sentence was removed from the manuscript according to comment #1 (major comments) and reviewer#2 suggestion.*

Reviewer: 4. Line 85, are there still amoeba cells alive at 48 hours post-infection? Why between 24 hours and 48 hours Spuntik and Zamilon do not continue to replicate (figure 1)?

ANSWER: *Yes, some amoeba cells are usually alive at H24 p.i., especially in case of Sputnik as it is a virulent virophage (PMID: 18690211). But DNA replication is a relatively early step in the virophage cycle. Indeed, in both virophage and giant virus cycles, the genome replication and morphogenesis steps do not occur concurrently. So, first the DNA is replicated and only at the end of this process that the morphogenesis begins. Because that, usually, there is no increase in virophage DNA amount after H24. However, in our experiments, we always prefer to check at 48h p.i. in case more than one replicative cycle takes place in the system (infection and reinfection).*

Reviewer: 5. Line 92, the authors say that the obtained supernatants containing only Guarani particles but, they do not explain how they discard the giant virus in this preparation (neither in materials and methods). Nothing is said concerning the MOI used.

ANSWER: *This has been explained just before by the sentence "After the lysis of host cells coinfecting with Guarani and Tupanvirus Deep Ocean or Tupanvirus Soda Lake, each culture supernatant was filtered through 0.22- μ m-pore filters to remove giant virus particles". Please note that Tupanvirus virions are giant particles that do not pass the 0.22- μ m-pore filter. The MOI was noted in the material and methods, and now also in the results according to the reviewer suggestion.*

Reviewer: 6. Line 122, Guarani genome consists of a circular dsDNA. Circularity was not experimentally proved. Remove circular

ANSWER: *This was removed as suggested.*

Reviewer: 7. Line 130, Table S1 does not exist

ANSWER: *This has been added-thank you.*

Reviewer: 8. In Figure 3b (right panel), the authors show a PCR product coming from Guarani produced with Tupanvirus, but they do not indicate which Tupanvirus they are using here. It will be helpful to see the PCR product coming from both Tupanviruses used as they say, in lines 130 and 131, that this mutation appears during the coinfection with both Tupanviruses

ANSWER: *As stated in the figure text, the PCR product shown in Figure 3 (Fig. 2 in the revised version) was done with Tupanvirus Deep Ocean. The same was observed for Tupan Soda Lake but, unfortunately, the quality of the gel that we possess in which TSL was tested simultaneously with TDO and APMV is not quite good to be presented for the manuscript. However, according to reviewer suggestion, the sequences obtained from each virus were attached in Supplementary data.*

Reviewer: 9. Line 137, the authors say that the sequencing of the PCR product allow them to localize the mutation in the ORF8 of the virophage, but that is not possible, because the target of the PCR is already the ORF8 so, they cannot extract this information from this PCR. They can only conclude that the deletion is present. Please, remove this sentence

ANSWER: *We agree. The sentence was removed-thank you.*

Reviewer: 10. Figure 4c, what is the MOI corresponding to 100 μ L of the mixture of WT and mutant Guarani?

ANSWER: *The MOI of virophage used at the beginning was 10 but then passages were performed blindly as explained above. The MOI corresponding to 100 μ l was not determined at each passage, it can vary from one passage to another.*

Reviewer: 11. Line 168, add the word giant in the sentence 'viable virus particles'

ANSWER: *Added- thank you.*

Reviewer: 12. Line 174, 'mutation increased its virulence', but, comparing with what? In this experiment there is no comparison, so the mutation cannot increase its virulence

ANSWER: *We completely agree, and the speculative sentence was modified.*

Reviewer: 13. Line 191, 'that mutant Guarani induces a severe decrease in the production of Tupanvirus virions'. Change production by infectivity (end-point dilution assay measure the infectious particles no the total number of particles)

ANSWER: *Modified as suggested-thank you.*

Reviewer: 14. Figure 5f-h shows always the same cell. Could be possible to see more cells instead of two different zooms?

ANSWER: *Yes, we agree, showing more cells is more representative, so, this has been added to the figure-thank you.*

Reviewer: 15. Line 209 and text figure 6, indicate the times post-infection. Can the authors discussed the difference of replication rates observed with the WT Guarani on APMV (reference 24 figure 1) and the mutant Guarani on TDO?

ANSWER: *Times post-infections were indicated Line 231. We agree that fitness of WT Guarani on APMV is significantly lower than that of mutant on TDO. This could be related to the adaptation of Tupanvirus transcription machinery to mutant Guarani genome. However, only extensive bioinformatic analyses could confirm this hypothesis and provides further explication. At this step, the results presented in this study do not allow us to propose any credible scenario regarding the fitness of the virophage in these two different giant viruses. This was commented in the revised version Line 313.*

Reviewer: 16. Line 219, add giant before 'virus'

ANSWER: *Added- thank you.*

Reviewer: 17. Line 244, add reference to Figure 7c

ANSWER: *Added- thank you.*

Reviewer: 18. The authors say that a subpopulation of Sputnik also showed the same deletion as the mutant Guarani (line 344). Unfortunately, the authors never used the mixture of WT and mutant Sputnik as a comparative virophage in their study. They preferred to use Zamilon, a virophage known to have no effect in giant viruses replication so far. So, next question arises, how different are Guarani and Sputnik during Tupanvirus infection? Comparative study would be very interesting

ANSWER: *We agree with this comment. But at the end, Guarani is a Sputnik strain and the ORF8 is the same between Guarani and Sputnik. As cited by reviewer 1 above, manipulating both virophages during a lot of experiences, including passage experiment, may be a source of cross contamination. Zamilon was used as a negative control as it is different and do not contain the same mutation. Otherwise, comparative study is interesting as precised in the perspective of the revised manuscript. We will try to investigate this in further studies. Thank you.*

Reviewer: 19. Line 447, how was the quantification of the TDO and TSL done?

ANSWER: *By end point dilution and calculated by the Reed–Muench method as done in our previous study. This was indicated in the materials and methods of the revised version.*

Reviewer: 20. Line 453, were the 200 µL aliquot taken from the supernatant or cells were taken and lysed to do the qPCR?

ANSWER: *From the supernatant, as indicated in the materials and methods of the revised version.*

Reviewer: 21. Line 515, add that, before doing the end-point dilution, the viruses were heating at 55°C, 30 min

ANSWER: *Added- thank you.*

Reviewer: 22. Line 520, how was the Guarani purified?

ANSWER: *The purification was explained in the materials and methods Line 504 – 512, as done in our previous work (PMID: 31130943).*

Reviewer: 23. Figure 9, add the MOI of virophage in the figure text

ANSWER: *Added- thank you.*

Reviewer: 24. The panels in the text of figure 10 are not well indicated

ANSWER: *Corrected-thank you.*

Reviewer: 25. Panel c of figure 10, authors are not using a pure mutant Guarani if not a mixture of WT and mutant, in this panel only the mutant is shown

ANSWER: *Corrected-thank you.*

Reviewer: 26. In future works, the authors should try to use the same infection condition (same MOIs in the different experiments) at least that the use of different conditions is justified to arrive at a conclusion. In this paper, different experiment are not comparable and make the understanding difficult.

ANSWER: *The system of tripartite virophage-giant virus-host cell is complex in itself. In this study, another variable, the mutation, has been added to this system, making it more and more complex. In our experiments, we have explored the interactions within this system by using three main technics: PCR, electronic microscopy and infectivity assay. Although, these methods have drawn to the same conclusion, as explained above, they do not measure the same parameters. This make the comparison between the results difficult as stated by the reviewer. Indeed, the virophage was monitored by PCR, which targets both infectious and noninfectious particles, in addition to free-released DNA. Moreover, the qPCR system used here also does not discriminate between mutant and WT Guarani, so, the background WT should also be taken into account when interpreting the results. On the other hand, the monitoring of the giant virus was done by two different methods: transmission electronic microscopy and infectivity assay. It is true that transmission electronic microscopy detects both infectious and noninfectious virions but the quantification using this technic was done manually by observing and counting a number of host cells and their virions. Hence, it may not reflect the exact proportions that exists in the system. In contrast, the infectivity assay targets only infectious virions. Despite the differences in what we were objectifying, these technics arrived at the one major conclusion; the mutant Guarani abolishes Tupanvirus propagation. We believe that this finding is in itself encouraging for a first identification and a preliminary characterization of virophage cross-species infection between giant viruses from different genera. This finding may have relevant application in understanding the interaction of the tripartite system in nature by showing that this interaction is more complex than previously thought.*

In this work, we tried to respect a defined chronology in performing experiments and interpreting the data. We believe that this will help reader in following the manuscript and understanding the information that each figure gives. At least, some experiments were carried out in different conditions in order to explore a specific parameter. This is the case for figures 3a and 3c (revised version) that do not have to be compared as their conditions are different. On the other hand, in experiments of Figure 6 (revised version), both giant virus virophage and host cells were studied using the same condition and during the one growth curve confirming the preliminary conclusions drawn from the passage experiments.

Point-by-point response to Reviewer #2 comments

Reviewer: As we know, defense systems in bacteria have been identified and demonstrated rather clear, however, it is still unclear in viruses, we knew that virus are specific for cognate host. This manuscript presented by Mougari S, et al., which reported virophage could abolish giant virus production and rescue the host cell population from virus lysis. The story presented seems well rounded, particularly; it should be interested in this field. Some issues will be addressed as following.

ANSWER: *Dear peer reviewer, thank you for your comments, we tried to follow all your recommendations.*

Reviewer: Major concerns: The manuscript describes that the deletion of ORF8 involved in integrating Guarani into the genome of Tupanvirus to allow the virophage to replicate. Here, there is no solid data to support this, if the authors could provide supportive data for this, the manuscript would be better.

ANSWER: *We completely agree that it would be very interesting if we could confirm this. Such data will significantly strengthen the impact of this manuscript. Unfortunately, as stated in the manuscript, for technical reasons, such experiment is not possible to set-up at this moment. However, the samples were prepared for sequencing and we will try to run them as soon as possible to incorporate them in the results of further studies. We have used Guarani isolated from passage 2 with Tupanvirus to infect fresh amoebas. The cells were then collected and conserved at 16h post infection, at this time, the virus factories were observed. We believe that it also remains to investigate several aspects of Guarani and Tupanvirus interaction. Namely, the role and the viability of the wild-type population present in the mixture obtained from Tupanvirus supernatant, and the capacity of the pure mutant virophage to replicate and cause Tupanvirus inhibition. For this, we will manage to use end-point dilutions coupled with PCR screening to clone the mixture and purify the mutant virophage. All these points were stated as perspective in the revised manuscript.*

Minor remarks.

Reviewer: 1/ Given how the study is, the introduction would be concise and focused for the reader to gain a basic understanding of host range expansion in a virophage.

ANSWER: *We tried to follow this recommendation by shortening some sentences in the introduction that discuss the mechanism of action of the MIMIVIRE defense system. The introduction discusses the notion of host range in virophage and we think that this is important to understand the manuscript. However, we apologize to not be able to concise the introduction more than this as reviewer #1 asked more information regarding the virophage Guarani and the giant viruses Tupanvirus Soda Lake and Tupanvirus Deep Ocean.*

Reviewer: 2/ lines142-144 could be modified the sentence for clearance.

ANSWER: *This has been modified-thank you.*

Reviewer: 3/ lines177-180 Could be consolidated into more single statement.

ANSWER: *Modified as suggested-thank you.*

Reviewer: 4/ Other statements could be clear and short, for example, lines300-323.

ANSWER: *Modified as suggested-thank you.*

Reviewer: 5/ Could be given the tittle of each figures.

ANSWER: *The organization of the figures has been modified in the revised version of the manuscript. Most figures have new titles. The list of the figures with their titles and their legends is indicated at the end of the manuscript.*

REVIEWERS' COMMENTS:

Reviewer #1 (Remarks to the Author):

I would like to thank the effort made by the authors to address all my comments which, from my point of view, has significantly improved the quality their manuscript.

Reviewer #2 (Remarks to the Author):

The revised version is better in well-writing and orgnization, etc. Almost questions have been corrected and modified, although my major concern is still unanswered. Certainly, I understand that the technical experiment needs long time to set-up. Overall, the current report is novel and interesting in microbiological field, so I suggest the Journal would be published it.